# Ultraflexible, cost-effective and scalable polymer-based phase change composites via chemical cross-linking for wearable thermal management

Yaoge Jing [1,4], Zhengchuang Zhao[2,4], Xiaoling Cao [1], Qinrong Sun[3], Yanping Yuan [1] ✉ & Tingxian Li [2] ✉

Phase change materials (PCMs) offer great potential for realizing zero-energy thermal management due to superior thermal storage and stable phase-change temperatures. However, liquid leakage and solid rigidity of PCMs are long-standing challenges for PCM-based wearable thermal regulation. Here, we report a facile and cost-effective chemical cross-linking strategy to develop ultraflexible polymer-based phase change composites with a dual 3D cross-linked network of olefin block copolymers (OBC) and styrene-ethylene-butylene-styrene (SEBS) in paraffin wax (PW). The C-C bond-enhanced OBC-SEBS networks synergistically improve the mechanical, thermal, and leakage-proof properties of PW@OBC-SEBS. Notably, the proposed peroxide-initiated chemical cross-linking method overcomes the limitations of conventional physical blending methods and thus can be applicable across diverse polymer matrices. We further demonstrate a portable and flexible PW@OBC-SEBS module that maintains a comfortable temperature range of 39–42 °C for personal thermotherapy. Our work provides a promising route to fabricate scalable polymer-based phase change composite for wearable thermal management.

Phase change materials (PCMs) are such a series of materials that exhibit excellent energy storage capacity and are able to store/release large amounts of latent heat at near-constant temperatures, making them play an indispensable role in thermal management technique innovation, e.g., temperature control techniques for flexible electronics like flexible photovoltaics, flexible batteries, and wearable electronics, which hold considerable potential to cope with environmental pollution and energy crisis[1–5]. Currently, organic solid-liquid PCMs (e.g., paraffin wax, fatty acids) have drawn great attention due to their stable physicochemical properties, low corrosivity and natural resources[6].

However, such conventional PCMs suffer from leakage issue, undesirable mechanical softness and toughness that cannot meet the requirements of poke, bend, twist, and stretch in thermal management applications of flexible electronics under complex environments[7,8]. To address the above issues of PCMs-based electronics and improve their broad-scale applicability, enhancing flexibility to adapt to demanding installation conditions by introducing appropriate carriers with characteristic structures inside PCMs is vital and highly desired.

Pioneer studies have reported that form-stable phase change materials (FSPCMs) obtained by embedding micro-molecular PCMs in

[1]School of Mechanical Engineering, Southwest Jiaotong University, Chengdu 610031, China. [2]Research Center of Solar Power & Refrigeration, School of Mechanical Engineering, Shanghai Jiao Tong University, Shanghai 200240, China. [3]School of Civil Engineering and Architecture, ChongQing University of Science and Technology, Chongqing 401331, China. [4]These authors contributed equally: Yaoge Jing, Zhengchuang Zhao. ✉e-mail: ypyuan@home.swjtu.edu.cn; Litx@sjtu.edu.cn

characteristic polymers or porous supporting matrixes are beneficial for suppressing irreversible damage caused by liquid leakage, e.g., container corrosion and environmental pollution[9–11]. Flexible PCMs typically rely on the structure of the support matrix to control the fluidity during phase change while maintaining a soft mechanical response, so that its applications in flexible electronics present the following advantages: (i) the brittle and fracture-prone rigid defects of conventional PCMs can be regulated[12]; (ii) the difficulty in processing and molding is reduced, so that various shapes can be made to meet the deformation requirements of equipment[13]; and (iii) better coordination with target device is supported, which is conducive to reduce contact thermal resistance[14]. In this regard, microencapsulation and/or shape-stabilization techniques have been extensively explored to achieve leakage resistance as well as the dominant special architectures[15]. Whereas the mainstream microencapsulation technique with a core-shell-like structure involved requires complicated fabrication procedure and easily sacrifices the energy storage density of PCMs due to excessive introduction of additives, which means difficult and high cost[16]. Another technique is to meet the flexibility requirements of PCM-based composites by employing suitable structural substances even at a low content loading. To this end, ceramic (e.g., silica aerogels, boron nitride aerogels)[14,17], carbon (e.g., graphene aerogels, carbon nanotube sponges)[18,19], fiber (e.g., hollow polypropylene fibers, cellulose fibers)[20,21], and thermoplastic elastomer (e.g., polystyrene, polyolefin and polyurethane)[12,22] based phase change composites have gained tremendous interests to achieve high energy density and exceptional flexibility. Among these substances, thermoplastic elastomers (TPEs) with similar microscopic phase separation structures, i.e., different resin segments (hard chains) and rubber segments (soft chains) composed of chemical bonds, are promising alternatives to flexible FSPCMs (F-FSPCMs). In especial, the dispersed hard chains of TPEs can form three-dimensional physical cross-linking points to be shaped by interchain force, while continuous soft chains contribute to flexibility and elasticity. Furthermore, it endows the following potentials: (i) better compatibility and homogeneity with organic PCMs than inorganic materials, (ii) less prone to collapse or phase separation than carbon-based materials, and (iii) a simpler fabrication technology and lower cost compared with fiber-based materials. Consequently, TPEs-based F-FSPCMs is an optimal technique for thermal management of flexible electronics from the perspective of low cost, portability, and human comfort, where the key challenge lies in the shaped matrix materials.

Note that styrene-ethylene-butylene-styrene (SEBS) consisting of poly-styrene (PS) resin segments and ethylene-co-butylene (EB) rubber segments can be regarded as a representative TPEs in terms of structure and characteristics. The leakage-proof property shown by SEBS and paraffin wax (PW) blends has been verified with PW loading up to 90 wt%[23,24]. Nonetheless, the rigid state of the SEBS/PW composite below the phase transition temperature and its highly elastic-gel state above the phase change temperature make the composite surface easily wrinkled after cooling, which negatively increases the contact thermal resistance of the bonding surface. Blending multiple TPEs allows the block copolymer morphology to be adjusted to achieve a modulation of the end-use properties. There are cases where SEBS compounded with low-density polyethylene (LDPE) or acrylonitrile-styrene-acrylate copolymer (ASA) as a co-matrix can further improve the performance deficiencies with only one TPEs[25,26]. However, the balance between the thermal and flexible properties of these blends is still non-ideal. A preferable method based on exploiting SEBS properties for efficient F-FSPCMs practice should be further explored. In another aspect, olefin block copolymer (OBC) with a unique multi-block structure has been demonstrated in relevant research to be effective as shaped matrix for PW, improving the shape stability, flexibility, and flatness[27,28]. Collectively, the flexibility enhancements of OBC in PCMs are still limited, leading to composites that are prone to

fracture when bent at temperatures higher than their phase transition temperature. Signs have indicated that the incorporated advantages of OBC and SEBS are beneficial for improving the elastic behavior of its blends to some extent[29]. Whereas the early explored PW-OBC-SEBS blends with low PW content focus on shape memory effects and the phase change properties are substantially weakened[30]. Besides, the involved fabrication is very dependent on harsh operating conditions (e.g., high temperature, high pressure, and high-speed mixing rates) governed by expensive heavy equipment (e.g., internal mixers, twin-screw extruders), which restricts large-scale production and the effect for thermal management is scarcer[31]. Recently, the preparation of functional materials (e.g., elastic ferroelectric materials) via cross-linking toughening to meet the performance enhancement for flexible wearable devices has gradually attracted attention, and such effect induced by C-C bond generation between polymers is worth exploring and utilizing in aforesaid polymer-based phase change materials[32–34].

Herein, we develop a cost-effective chemical cross-linking method to synthesize leakage-proof, ultraflexible, and thermally insulating polymer-based phase change composites for thermal management of flexible electronics. PW with high phase-transition enthalpy and low-cost natural resource as the energy storage medium is immobilized within the formed three-dimensional (3D) cross-linking networks of SEBS-OBC. As a note, an atmospheric low-speed sequential melt blending method, coupled with peroxide-induced chemical cross-linking effects, allows for the efficient batch and rapid fabrication of PW@OBC-SEBS composites (i.e., F-FSPCMs here). By virtue of the unique interplay of OBC-SEBS on brittleness and elasticity improvement, the F-FSPCMs display substantially enhanced flexibility, with the breaking strain increasing from 23% to 560%. Furthermore, as a proof of concept, an integrated and portable F-FSPCMs module is well designed and a constant temperature effect (39–42 °C for 43 min) during repeated cycling is achieved, thereby demonstrating the free-standing temperature control capability for wearable thermal management, which beneficially improves personal thermal comfort. In addition, F-FSPCMs also exhibit high latent heat enthalpies up to 176.0 J/g, excellent thermal stability, and strong hydrophobicity, indicating their potential utility for high-power-density applications in wearable thermal management of flexible electronics.

## Results
### Strategy of F-FSPCMs
Figure 1 depicts the schematic diagram for fabricating PW@OBC-SEBS composites, also referred to as F-FSPCMs. Given the strong compatibility between SEBS and PW, we adopted a sequential melt blending approach to first disperse SEBS in PW, modifying them to PW-SEBS gel. This process was carried out in a beaker at a stirring speed of 200 rpm. OBC was then melted into the gel by reducing the stirring rate to 100 rpm. Subsequently, Di(tert-butylperoxyisopropyl)benzene (Peroxide) was added to the composites, leading to the generation of numerous tiny and dense bubbles. These bubbles gradually disappeared after about 30 min as the reaction proceeded, giving homogeneous cross-linked products. All these processes were carried out at 180 °C and atmospheric pressure. Finally, PW@OBC-SEBS composites were obtained by cooling the mixture in a homemade mold. By scaling up the above experimental feeding to the equipment equivalently, we can obtain 20 kg of F-FSPCMs in a single preparation (For more details, see Supplementary Figs. S1–S3, Table S1 and the "preparation method" section).

Specially, the cross-linking reaction of F-FSPCMs in liquid PW has three steps. The first step involves the labile oxygen-oxygen bond of the organic peroxide undergoing homolytic cleavage to form primary alkoxy radicals, which is the rate-determining step of the overall response. The second step involves the radicals extracting hydrogen atoms from the elastomer chains to produce stable gaseous peroxide decomposition products and macroradicals of the SEBS and OBC

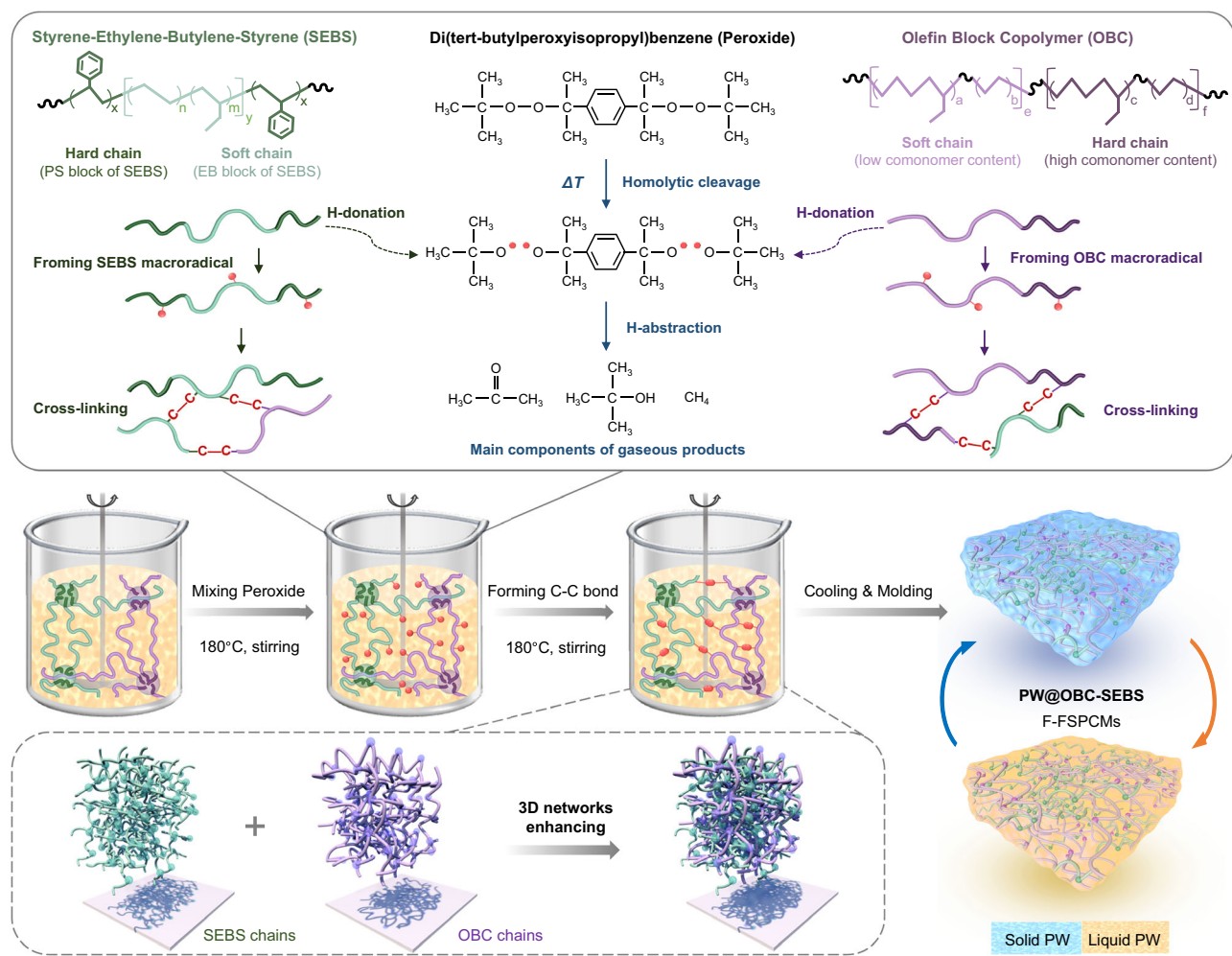

**Fig. 1 | Illustration of the procedure for fabricating PW@OBC-SEBS composites (F-FSPCMs).** Peroxide induces homolytic cleavage, generating free radicals and promoting C-C bond formation, enhancing OBC-SEBS networks and leading to PW@OBC-SEBS after molding.

elastomers. At this stage, the formation of radical centers on the elastomer chains enhances the activity of the OBC and SEBS molecular chain segments, thus speeding up the homogeneous mixing process and saving preparation time. The last step involves the recombination of OBC macroradicals and SEBS macroradicals to generate carbon-carbon (C-C) bonds. The full release of the gas products marks the formation of enhanced dual 3D network F-FSPCMs.

Details of the prepared samples are listed in Table 1. The host material PW allows for the retention of high latent heat enthalpy of the composites, while the guest materials OBC and SEBS are expected to endow the composites with leakage-proof properties and ultra-flexibility.

## Table 1 | Test F-FSPCMs schemes

| Samples-content of peroxide (wt%) | Peroxide (g) | OBC (g) | SEBS (g) | OBC/SEBS (mass ratio) | PW (g) |
|---|---|---|---|---|---|
| S0 (0.67) | 0.5 | 15 | 0 | 1/0 | 60 |
| S1 (0.67) | 0.5 | 12.5 | 2.5 | 5/1 | 60 |
| S2 (0.67) | 0.5 | 11.25 | 3.75 | 3/1 | 60 |
| S3 (0.67) | 0.5 | 7.5 | 7.5 | 1/1 | 60 |
| S2-0.0 | 0 | 11.25 | 3.75 | 3/1 | 60 |
| S2-1.33 | 1 | 11.25 | 3.75 | 3/1 | 60 |
| S2-2.0 | 1.5 | 11.25 | 3.75 | 3/1 | 60 |

## Cross-linking and characterization of F-FSPCMs

To validate the composition of gas bubbles produced during the experiment, we collected gas samples near the mouth of the container using a gas bag in the preparation process. The collected gases are then analyzed for methane and acetone content using gas chromatography (GC) and gas chromatography-mass spectrometry (GC-MS). As depicted in Fig. 2a, b, the levels of methane and acetone associated with the chemically crosslinked PW@OBC-SEBS are detected at 54.2 mg/m³ and 2096.0 μg/m³, respectively. This represents an ~30-fold increase compared to the levels found in physical blended PW-OBC-SEBS. It is noteworthy that peroxide only generate decomposition products upon abstracting hydrogen atoms from the polymer, thereby aligning with the gas species predicted in the theoretical analysis (seen in Supplementary Fig. S1 and Table S1).

Raman spectroscopy appears to be a promising technique for determining the degree of PW@OBC-SEBS cross-linking[35,36]. Figure 2c displays characteristic fingerprint regions corresponding to the C-C stretching and C-H stretching vibrations. The Raman spectra exhibit bands at 1000–1200 cm⁻¹ (C-C stretching vibration), 1295–1305 cm⁻¹ (CH₂ twisting), 1442 and 1460 cm⁻¹ as a shoulder (CH₂ deformation vibrations), 2850 and 2883 cm⁻¹ (aliphatic CH₂ symmetric and asymmetric stretching vibrations), respectively[37,38]. The highest bands' intensity at 2883 cm⁻¹ is utilized to normalize the Raman spectra in the 800-4000 cm⁻¹.

When the OBC/SEBS ratio changes, the peak intensities of samples S0-S3 at 1060, 1132, 1295, 1442, 1460, and 2850 cm⁻¹ show little

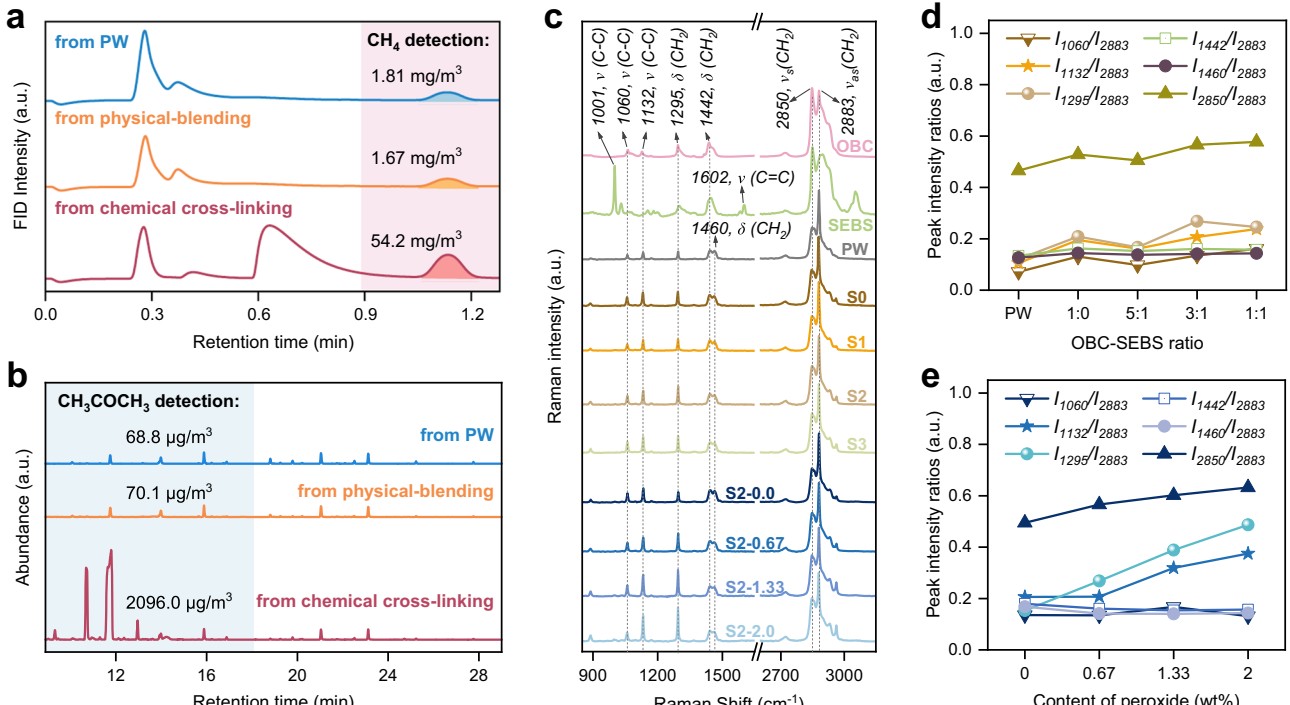

**Fig. 2 | Demonstration of F-SPCMs cross-linking response. a** GC response curve for methane detection. **b** Total ion chromatogram of acetone detected by GC-MS. **c** Raman spectra of OBC, SEBS, PW, and all fabricated samples. **d** Evolution of the relative Raman intensities for F-FSPCMs with different OBC/SEBS ratios. **e** Evolution of the relative Raman intensities for sample S2 with increasing content of peroxide.

fluctuation and differ little from the peak intensity of PW (Fig. 2d). Whereas with the peroxide content increasing from 0 to 2.0 wt%, the peak intensities of F-FSPCMs with a fixed OBC/SEBS ratio double in the bands 1295 and 1132 cm⁻¹ (Fig. 2e).

The symmetric stretching strength of the C-C bond at 1132 cm⁻¹ increases, associated with the trans conformation, while 1295 cm⁻¹ indicates a twisted vibration of CH₂, related to the trans segments and gauche conformers. Generally, macromolecules in the crystallites (orthorhombic crystalline phase) are in the all-trans conformation. The non-crystalline regions (interphase and liquid-like amorphous phase) contain both macromolecules in the trans conformation and gauche conformers[39]. Hence, cross-linking appears to occur more frequently in the non-crystalline region, in the case of OBC and SEBS, which consists of polyethylene and alpha-olefin monomers (mainly 1-octene and 1-butylene) cross-linking with each other (Supplementary Fig. S2). Due to the formation of new C-C bonds, the rotation within the bond results in a different spatial arrangement of the atomic groups in the molecule, leading to conformational isomerism and an increase in the intensity of the CH₂ twisting at 1295 cm⁻¹.

Furthermore, the joint verifications of SEM and AFM (Supplementary Fig. S5 and S6) indicate that the formulation of polymer OBC/SEBS in the presence of peroxide cross-linking can lower the surface roughness of phase change composite and help reduce the contact thermal resistance for material practice, providing a basis for the heat transfer characteristics in future F-FSPCMs and electronic devices applications. A detailed discussion on the crystal morphology of F-FSPCMs including polarizing optical microscope (POM), FT-IR and XRD results (Supplementary Fig. S7-S9) further imply that the crystallization of PW is constrained by the 3D network structure of OBC-SEBS, with OBC showing a dominant effect on the crystallization in combination with the following DSC results.

## Thermal performance of F-SPCMs

The phase change performance of F-FSPCMs is analyzed by DSC, involving heating and cooling processes in both low and high-

temperature regions (Fig. 3a). The phase transitions of PW and OBC are observed in the low and high-temperature regions, respectively, whereas the curve of SEBS is not analyzed since no phase transition occurred within this temperature range. Notably, a portion of PW enters the crystallizable chain segments of the OBC, which leads to a change in the heat absorption peak of the OBC from a characteristic single peak to a double peak, similar to that of PW. Nevertheless, the introduction of SEBS alleviates the effect of OBC on PW crystallization. This contrasts with previous reports that PW permeates in OBC and SEBS exhibits separated melting transitions, confirming that the higher content of PW herein minimizes the unexpected effect of OBC-SEBS on PW crystallization[30].

Thermal parameters including phase change temperature and enthalpy are extracted and summarized in Fig. 3b and Table S2 (Supplementary). The definition specification of $T_m$ and $T_f$ can be found in Supplementary Information (Fig. S10). The curves of samples S0-S3 with identical content of PW are very close despite the difference in OBC-SEBS ratios. Where the phase change temperature of all cross-linked samples is $42 \pm 0.5\,°C$ without significant difference, and the enthalpy is as high as 176.0 J/g, with a minimum decrease of 6.6 J/g compared with the theoretical value (calculated by equation $\Delta H_{\text{F-FSPCMs}} = \varphi_{\text{PW}}\Delta H_{\text{PW}}$)[40]. These results highlight that the phase change behaviors of F-FSPCMs are dominated by the host material PW. Apart from characterizing thermal performance, DSC is also a promising alternative for measuring cross-linking degree. With increasing peroxide concentration from 0 wt% to 2.0 wt%, the melting enthalpy of F-FSPCMs (S2) at low temperatures decreased from 177.1 J/g to 157.5 J/g, while the crystallization temperature at high temperatures decreased from 99.6 °C to 97.0 °C. Such situation is due to the difficulty in the regular molecular chain arrangement and the retardation of crystal growth as the cross-linking density increases. This drives a decrease in the thickness and size of the crystals, leading to a reduction in the enthalpy and phase transition temperature of F-FSPCMs.

Thermal cyclability and thermal decomposition processes of F-FSPCMs are significant references for evaluating the thermal stability

between the components of F-FSPCMs. As shown in Fig. 3c, after experiencing 500 accelerated thermal cycles, the thermal properties of PW and OBC in F-FSPCMs scarcely change compared to their original state, indicating the excellent thermal stability of F-FSPCMs for long-term application. From the TGA results in Fig. 3d, the decomposition of pure PW, OBC, and SEBS occurs within the temperature ranges of 152–266 °C, 400–490 °C, and 387–470 °C, respectively. For F-FSPCMs, its thermal decomposition can be divided into two steps, with the first step occurring at 200–290 °C and a mass loss of 80% (exactly the mass fraction of PW), and the second step occurring at 430–480 °C corresponding to the decomposition of OBC and SEBS. Besides, the thermal decomposition differences between slightly crosslinked PW@OBC-SEBS with different proportions are minimal. It is worth emphasizing that the OBC-SEBS chain network structure inside F-FSPCMs is deemed to protect PW from evaporation to some degree, thus increasing its onset decomposition point by 50 °C. Therefore, our fabricated F-FSPCMs possess better heat resistance and thermal stability than pristine PW. Furthermore, incorporating a dual three-dimensional cross-linking network of OBC-SEBS into the PCM still maintains a relatively low thermal conductivity for F-FSPCMs (seen in Fig. 3e). The thermal conductivity of F-FSPCMs remains within the range of 0.23–0.27 W m$^{-1}$ K$^{-1}$ before the phase transition (at 35–40 °C), and experiences a slight decrease to 0.16 W m$^{-1}$ K$^{-1}$ after the phase transition (>45 °C), thus contributing to the effective thermal insulation for personal thermal management.

## Flexibility and leakage-proof properties of F-FSPCMs

Flexibility is a critical characteristic for PCMs in thermal energy storage and management applications, as it denotes the material's ability to deform without breaking under external forces. The superior flexibility of the as-fabricated F-FSPCMs is demonstrated by their bending process (Fig. 4). As the temperature increases, the PW changes from a solid to a liquid state, while the soft chain segments of OBC-SEBS transition from a frozen state to a free-moving or rotating state, rendering the

composites soft and pliable. Simultaneously, the gaps between the segments are filled with liquid PW but without leakage, since the hard chain segments of OBC-SEBS remain cross-linked. Furthermore, the F-FSPCMs can withstand external forces such as curling and bending, which reduces the density of the stretched chain segments and increases the density of the extruded chain segments. The chain segments of OBC-SEBS can be temporarily immobilized when the PW changes from a liquid to a solid state. Upon reheating, the F-FSPCMs slowly revert to their initial shape and quickly regain their shape aided by an external force. After cooling below the phase change temperature, the composite's shape, rigidity, and hardness return to their original states.

To further validate the mechanical characteristics of F-FSPCMs, mechanical tensile and compressive assessments are performed on F-FSPCMs at different temperatures. Figure 5a depicts stress-strain profiles and schematic illustrations elucidating the tensile testing procedure. The results clearly indicate that the F-FSPCMs exhibit reduced peak stress and modulus as well as increased strain with increasing temperature (Fig. 5b–d). Notably, significant differences in strain at break are observed for S0-S3 at the same temperatures. Specifically, SEBS content can enhance ductility significantly, the strain at break can be increased from 75% to 560% at 46 °C. On the other hand, as the peroxide content increases from 0 wt% to 2 wt%, there is an initial increase followed by a decrease in strain at break, which was consistently observed across all temperatures due to the heightened cross-linking density. A low cross-linking density positively impacts network structure performance, while excessive cross-linking density would in turn restrict molecular chain slippage, consequently limiting the mechanical properties of the blend. This observation aligns with established principles in previous researches, indicating the existence of an optimal degree of mild cross-linking that can be used to regulate fracture elongation to achieve desired values[33,41].

Figure 5e presents a comprehensive comparison of the maximum latent heat and degree of flexibility achievable by various typical

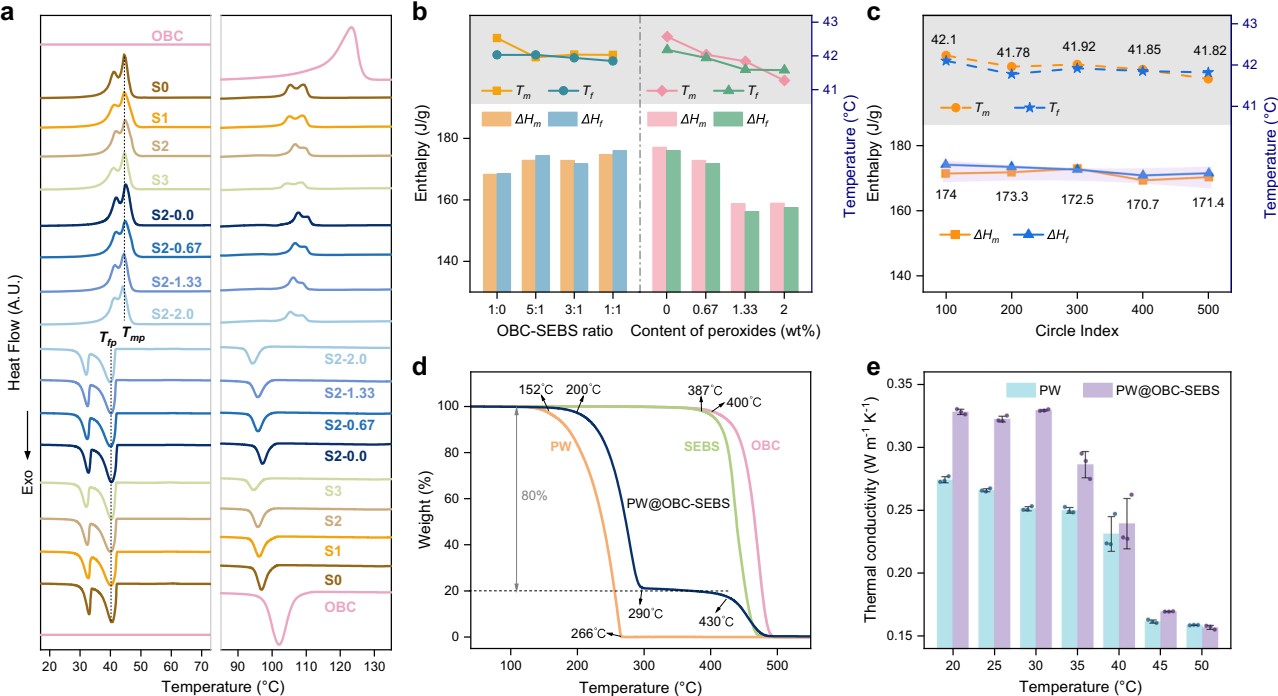

**Fig. 3 | Thermal properties of PW, OBC, and PW@OBC-SEBS composites (F-FSPCMs). a** DSC curves of OBC and all fabricated samples. **b** Enthalpy and phase change temperature of all fabricated samples. **c** Enthalpy and phase change temperature of PW@OBC-SEBS (S2) after 500 accelerated thermal cycles. **d** TGA curves

of PW, OBC, SEBS, PW@OBC-SEBS (S2). **e** Thermal conductivity and diffusivity comparison of PW and PW@OBC-SEBS (S2). Error bars in **e** are standard deviation (s.d.) from 3 samples.

shaped matrixes[24–26,40,42–53]. Encouragingly, our F-FSPCMs exhibit higher enthalpies and stretchability and have breaking strain tolerance of 15-133% without fracture or damage below the phase change temperature, while the breaking strain can also steadily increase from 69% to 156% or 144% to 560% near or above the phase change temperature, respectively.

The F-FSPCMs exhibit distinct mechanical behaviors before and after the phase change under compression, as depicted in Fig. 5f, g. At a temperature below phase change point, increasing SEBS content can improve the compressive strength continuous deformation of F-FSPCMs during the compression process, eventually resulting in a flattened rectangular prism shape. However, at a temperature above phase change point, the F-FSPCMs manifest internal cracks and defects as the compressive strain approaches nearly 20% when the OBC/SEBS ratio surpasses 3:1, leading to a decrease in peak compressive stress. Conversely, at ratios of 3:1 and 1:1, the F-FSPCMs exhibit similar elastomeric characteristics with superior compressive strength. Moreover, after undergoing 200 cycles of 30% compression fatigue testing, the F-FSPCMs consistently followed the same stress-strain curve during both loading and unloading processes (Fig. 5h), making them a promising choice for thermotherapy module applications that demonstrate substantial deformability and high stability under repetitive compression.

Notably, as the degree of chemical cross-linking increases below the phase transition temperature, both compressive and tensile modes gradually decrease peak stress and elastic modulus. Conversely, there are no significant changes above the phase transition temperature (46 °C) (Supplementary Fig. S11). The higher tensile stress and modulus are primarily related to the degree of entanglement of central chain segments within the elastomer[54]. During the cross-linking process, OBC-SEBS molecular segments undergo dynamic fracture and recombination, forming new C-C single bonds and reducing the polymer's average molecular weight, leading to a relative decrease in stress intensity and modulus. As the degree of cross-linking increases, cross-linking reactions become more pronounced and molecular segments tend to experience more significant disruption, ultimately reducing their mechanical properties. Therefore, it should be emphasized that the regulatory mechanism of "chemical cross-linking" does not follow a fixed pattern, especially considering that the paraffin content in our phase change composites accounts for 80% rather than simply a single thermoplastic elastomer cross-linking system.

Lower elastic modulus and higher strain at break can be regarded as indication of flexibility for our F-FSPCMs base on statements in cross-linking relevant researches[55,56]. Further, we observed similar characteristics in other material systems (PW@OBC, PW@POE-LDPE) (Supplementary Fig. S12 and Fig. S13). Specifically, chemical cross-

linking driven by peroxide decomposition reaction exhibits a substantial enhancement effect on flexibility compared to physical blending. This insight suggests that, beyond the choice of raw materials and their proportions, the utilization of peroxide cross-linking in thermoplastic elastomers or cost-effective plastics can serve as a universal means for engineering flexible phase change materials.

In terms of the intrinsic structure of the ternary components, F-FSPCMs are expected to possess hydrophobicity. To validate this, the water contact angle of PW@OBC-SEBS is measured to be 123.5°, indicating its potential for direct heat exchange with water (Supplementary Fig. S15a). Inspired by this, in contrast to conventional mild leakage testing methods, we performed a destructive experiment under harsh conditions to evaluate the leakage behavior of PW, aiming to maximize its precipitation. The implementation is shown in Fig. S15b. The results demonstrate that PW@OBC-SEBS exhibited better leakage-proof performance than PW@OBC, which is attributed to the synergistic network of dual 3D cross-linking dominated by the OBC-SEBS matrix, as well as the hydrophobicity and the potential capillary effect, contributing to enhance leakage-proof and form-stable abilities. Then, it can be anticipated that further studies based on this approach could provide solutions for energy storage temperature control in humid environments, mitigating issues such as decreased energy density, reduced mechanical strength, and bacterial growth caused by the hygroscopicity of hydrophilic support materials.

## Demonstration of F-FSPCMs for wearable thermal management

Thermal management is critical for electronic devices as excessively high or low temperatures can significantly impact their energy consumption and lifespan. In this study, we demonstrate the use of F-FSPCMs modules for temperature control, which utilize polyimide (PI) heating belts as heat sources and are inserted into the F-FSPCMs. Figure 6a displays the experimental setup and the schematic diagram of the heat storage/discharge, including thermocouple arrangement. Upon cooling and solidification in our module preparation, the two components, i.e., F-FSPCMs and PI heater band, can form a tight, air-free bond at their interface, and consequently, a flexible and contour-controlled phase change module is widely available. Further details are visible in the experimental section.

The temperature variations versus time indicates a consistent trend for all 8 test points (Fig. 6b), demonstrating decent temperature uniformity during heat storage/discharge cycles. Following power-off, a delayed occurrence of peak surface temperature during the charging phase should be observed, indicating a heat transfer delay due to the poor thermal conductivity of F-FSPCMs (Fig. 3e). But encouragingly, this situation is eliminated within 1 min under all conditions, and less noticeable as the module thickness becomes thinner (Supplementary

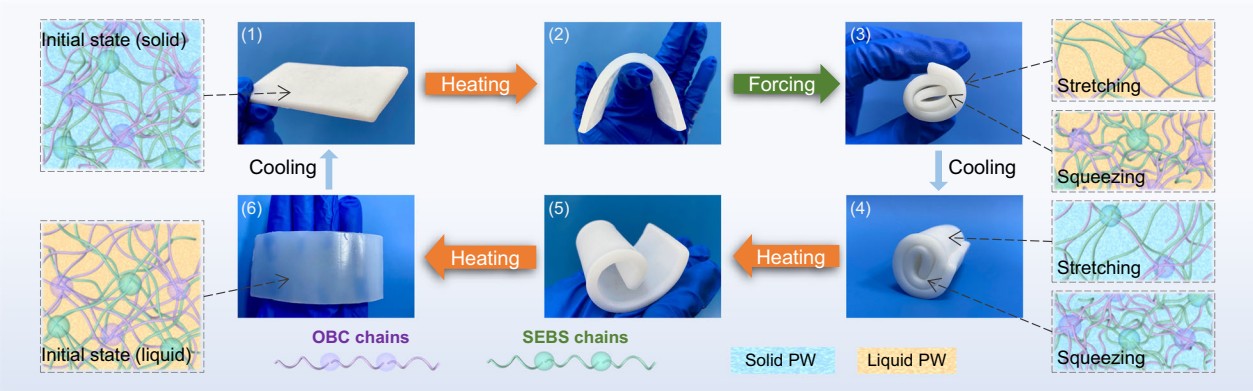

**Fig. 4 | Schematic illustration of the freeze-release of OBC and SEBS chain segments during bending of F-FSPCMs.** The process includes six stages: (1) initial state; (2) after heating; (3) after forcing; (4) after cooling; (5) continuing heating; (6) final state, recovering to the initial state after cooling.

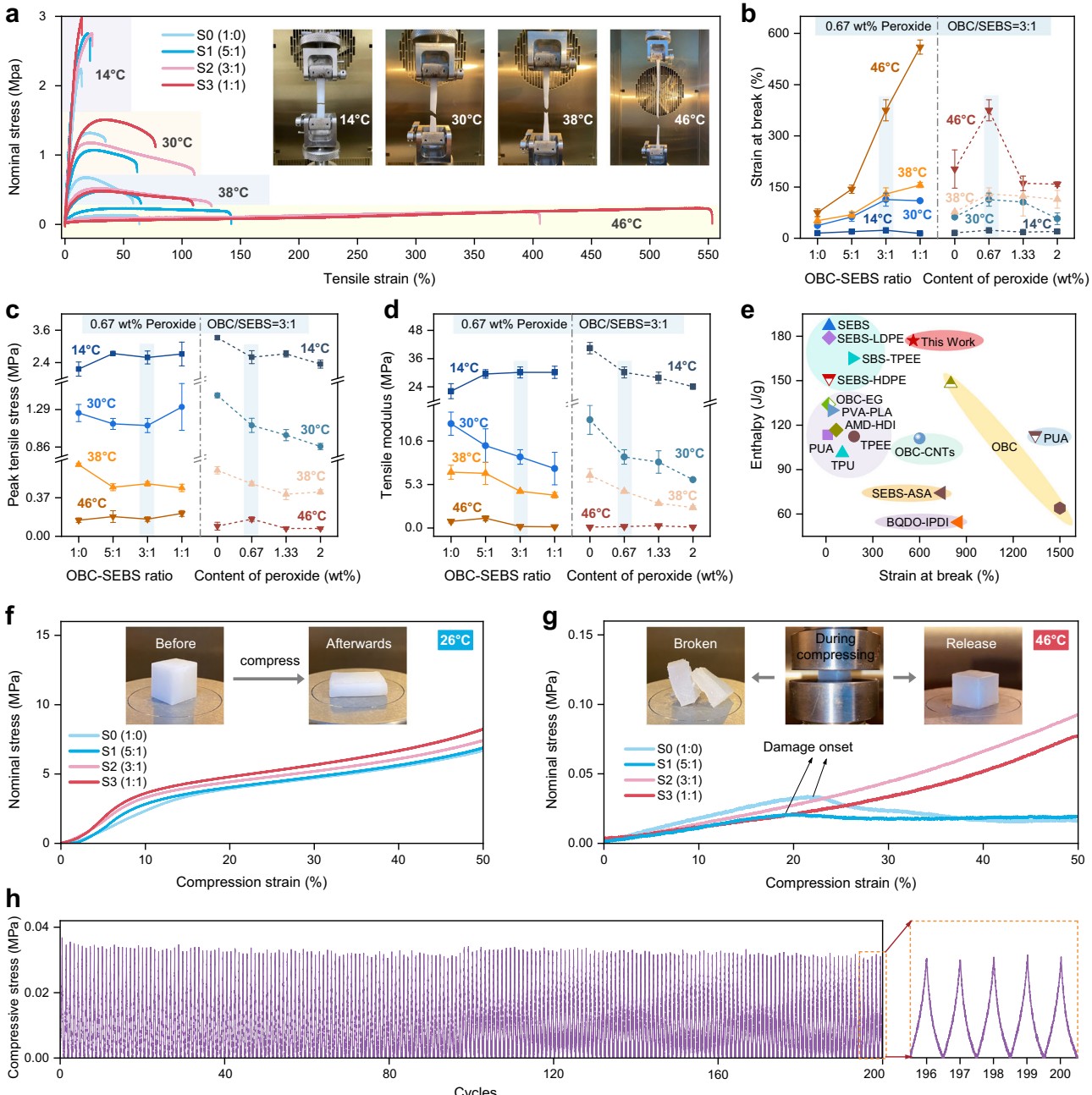

**Fig. 5 | Mechanical Properties of PW@OBC-SEBS composites (F-FSPCMs).**
**a** Tensile stress-strain curves of F-FSPCMs at different temperatures. **b** Influence of
OBC-SEBS ratio and peroxide concentration on strain at break. **c** Effect of OBC-
SEBS ratio and peroxide concentration on tensile peak stress. **d** Impact of OBC-
SEBS ratio and peroxide concentration on tensile modulus. **e** Comparison of strain
at break and enthalpy of our prepared F-FSPCMs with literature[24–26,40,42–53]. And the
detailed information is shown in Table S3 (Supplementary). **f** Compression stress-
strain curves of F-FSPCMs below the melting point. **g** Compression stress-strain
curves of F-FSPCMs above the melting point. **h** Cyclic compression performance of
F-FSPCMs (S2-0.67) at 46 °C compressed to 30% strain over 200 cycles. Error bars
in **b**–**d** are standard deviation (s.d.) from 3 samples.

Fig. S16). Furthermore, a color change module for F-FSPCMs is devel-
oped to indicate temperature variation, where the phase change
temperature is used to stimulate color change to achieve self-adapting
charging/discharging. Therefore, it avoids the frequent temperature
feedback for power on and off in the conventional power-heat supply
methods.

Moreover, Fig. 6b shows that during the discharging stage, the
first-stage of phase change latent heat release occurs from point C to
D with the surface temperature maintained in the interval of 39–42 °C
for 44 min. Then, after partial sensible heat release (from point D to E),
the second-stage of phase change latent heat release operates (from

point E to F), and the surface temperature can be maintained between
34 and 36 °C for 22 min. Where point F represents the end of the total
phase change. This outcome implies an outstanding two-stage tem-
perature control ability and high thermal efficiency potential of
F-FSPCMs. We further analyzed the effect of module thickness on the
two-stage constant surface temperature (Fig. 6c). The obtained results
provide a guideline for the design of modular specifications to meet
practice requirements. The calculated thermal efficiency of the
F-FSPCMs modules can reach up to 96.3% whilst accompanying by a
comfortable peak surface temperature of 43.3 °C (Supplementary
Table S4), showing a great potential for personal thermal

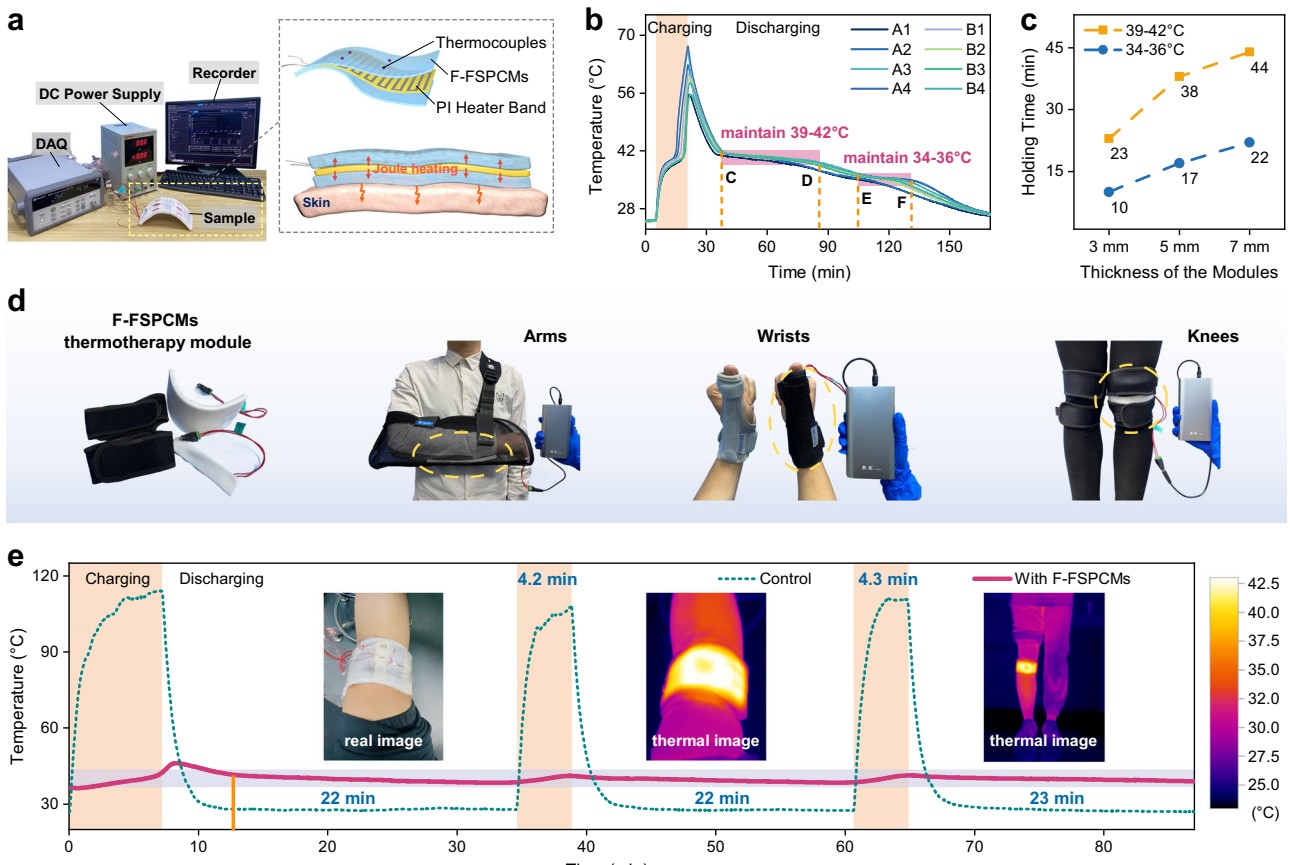

**Fig. 6 | Thermal management applications of PW@OBC-SEBS composites (F-FSPCMs). a** Experimental setup for the evaluation of heat storage/release of the F-FSPCMs module. **b** Surface temperature of the F-FSPCMs module during a heat storage/discharge cycle (7 mm thickness). **c** Temperature holding time of the F-FSPCMs module with the same heat supply for 3 mm, 5 mm and 7 mm thickness. **d** Several types of integrated and portable F-FSPCMs modules for personal thermotherapy. **e** Temperature evolution of insulating device based on F-FSPCMs module (5 mm thickness) acting on a knee during charging and discharging.

management, e.g., thermal buffering regulation under high heat flux density conditions.

Based on the aforesaid temperature control characteristics, an intermittent heating approach is also employed to obtain a more extended periodic constant temperature effect. In this experiment, when the temperature drops to the lowest point of the target temperature interval in the cooling exothermic stage, the power is re-energized and subsequently heated back to the highest point of the interval, so that the surface temperature is maintained at the phase transition temperature. Where the target temperature range is selected to be 39–42 °C. Typically, the temperature control effect of intermittent energized thermal storage of the 7 mm thickness F-FSPCMs module is tested at an ambient temperature of 25 ± 1 °C (Supplementary Fig. S17). The module demonstrates a practical effect of 5 min charging and 42-45 min of thermostability after the first heating and cooling cycle. Throughout the charging and discharging process, a total of 25 min of powering and 187 min of thermostability are achieved, ensuring the efficient bidirectional heat storage and long-term temperature control capability of the F-FSPCMs module for wearable thermal management.

By virtue of reducing the peak surface operating temperature, or rather excellent temperature control performance, we further apply the F-FSPCMs module on human knee as an example of thermotherapy to detect its applicability for personal thermal management. The flexibility enhancement and bioavailability driven by the OBC-SEBS matrix allows us to fabricate F-FSPCMs modules with a large specification, and which can be bent to fit human knee curvature without

leaking F-FSPCMs (left image in Fig. 6d). For ethical considerations, a temperature control module with a thickness of 5 mm is preferred to enable subjects to test multiple heating-cooling cycling behavior. The heat charge-release curve in Fig. 6e shows that a suitable temperature range of 39–42 °C for human body can be maintained for >20 min after the first charge. Moreover, the stagnation time corresponding to such thermal comfort temperature after the second and third short charge almost coincides with the first one. The IR thermal image of which also highlights that the charged F-FSPCMs module evenly releases heat to the knee, which enable F-FSPCMs modules to exhibit a superior thermal therapy effect, collectively demonstrating their better thermal cycling stability and long-term service lifespan in practical applications.

Furthermore, when combined with commonly used rehabilitation equipment, modules with controllable thickness can be fixed on different human body parts, including knees, fingers, wrists, arms, etc. As shown in Fig. 6d, several integrated and portable F-FSPCMs temperature control modules for personal thermotherapy have been designed with excellent fitting ability and wearability, allowing them to be flexibly bent, removed, and (or) adjusted. Compared with the split phase change material and heating device structure, the modular design form here greatly simplifies the structural process, reduces the weight, and shortens the manufacturing cycle. Importantly, the thermal charging of F-FSPCMs module is not limited to electrical energy, it can also be heated by low-grade waste heat or solar energy. Specially, it also has great potentials for battery thermal management, solar photothermal conversion and thermal storage.

## Discussion

In summary, we propose a cost-effective chemical cross-linking method to synthesize polymer-based phase change composites with ultraflexibility and high thermal storage density. The theoretical and experimental results contribute fresh insights into understanding the mechanism of chemical cross-linking effect induced by free radicals from peroxide decomposition. We systematically investigated the chemical cross-linking toughening effects of polymers under various proportions of OBC/SEBS and peroxide. The introduction of peroxide reinforces OBC-SEBS dual 3D networks for encapsulating paraffin, and thus enables the resultant PW@OBC-SEBS to exhibit high latent heat enthalpy (-176.0 J/g), reliable thermal stability, flexibility (strain at break of up to 560%, stable over 200 compression cycles), and leakage-proof performance. Importantly, the chemical cross-linking method can realize the high yields of F-FSPCMs from 100 g to 20 kg even in the laboratory under the peroxide-assisted sequential blending. Furthermore, we demonstrate a portable and flexible F-FSPCMs module for wearable thermal management. The energy module can maintain a stable and comfortable temperature range of 39–42 °C and achieve a discharging duration >22 min for personal thermotherapy after charging for 5 min. Our work provides a promising route to engineering scalable and flexible polymer-based phase change composites for high-power density thermal management, especially for electronic devices and wearable applications.

## Methods

### Materials

Paraffin wax (PW, Brand for OP44E) with a melting temperature of 41–44 °C as the base phase change material is supplied by RuhrTech (China). OBC (INFUSE™ 9530) and SEBS (FG1901, 30/70) preferred as polymer supporting materials owing to their unique molecular structures and good compatibility with PW are purchased from Dow Chemical Company (USA) and Kraton Corporation (USA), respectively. The density of OBC is 0.887 g/m³ and the melt flow rate is 5 g/10 min (2.16 kg at 190 °C). Di(tert-butylperoxyisopropyl)benzene (CAS:25155-25-3, $C_{20}H_{34}O_4$) is an organic peroxide of type Perkadox 14S-FL supplied by Akzo Nobel N.V. (Netherlands).

### Characterization and measurements

Methane production is quantified using a gas chromatograph (GC, SP3420A, bfrl, China) equipped with a Flame Ionization Detector (FID). The chromatographic conditions employ a GDX-104 column, with a column temperature of 50 °C, an injector temperature of 100 °C, and a detector temperature of 200 °C. The split ratio is set at 10:1, and the injection volume is 1.0 mL. Acetone concentration is measured using a gas chromatography-mass spectrometry system (GC-MS, 8890-5977B, Agilent, USA). A 400 mL sample is introduced into a cryogenic trap concentrator, and 50 mL of a standard gas for internal standard method is simultaneously injected. The conditions are set with a DB-1 column, an injector temperature of 150 °C, a column flow rate of 1 mL/min, and a split ratio of 20:1. The temperature program is initiated at 5 °C, held for 15 min, ramped at 8 °C/min to 180 °C, further increased at 10 °C/min to 220 °C, and then held for 9 min. The Raman spectroscopy (Lab RAM HREvolution, HORIBA, Japan) with a 532 nm laser is used to collect the Raman spectra. Three different spots are tested on the same sample at ambient temperature. Scanning electronic microscopy (SEM, JSM 7800 F Prime, Japan) is applied to observe the morphologies of the samples after gold spray treatment. Surface roughness of smooth samples after hot pressing is measured using the tapping mode of an atomic force microscope (AFM, Cypher S, OXFORD INSTRUMENTS, UK). The crystalline morphologies of samples are studied using a polarized optical microscope (POM, LEICA DM 2700 P, Germany), and a Linkam optical system (LTS420, UK) is used to control the temperature precisely. Samples are firstly heated to 120 °C and maintained at 120 °C for 5 min. Then samples are rapidly cooled down to relevant crystallization temperatures. Fourier transform infrared (FT-IR, iS50, Thermo Nicolet, USA) is used to investigate sample's chemical structure in the wavelength range of 4000 cm⁻¹-500 cm⁻¹. The crystal structure of the samples is investigated by X-ray diffraction (XRD, PANalytical Empyrean, Netherlands) at scanning 2θ ranging from 5° to 60°. The thermal properties of the samples are measured by using differential scanning calorimetry (DSC, TA Q20, USA) under a nitrogen atmosphere with a heating/cooling rate of 5 °C/min in the temperature range of 10–140 °C. All the samples are first heated from ambient temperature to 140 °C at 10 °C/min under a nitrogen atmosphere to erase the previous thermal history, and then they are cooled to 10 °C at the same rate. Moreover, the thermal reliability of the F-FSPCMs (S2) is investigated after 500 accelerated thermal cycles in a dry bath (Bioer Thermo Q, CHN) in the temperature range of 20–80 °C at a rate of 10 °C/min and maintained for 1 min at 20 and 80 °C during each thermal cycle. A thermal gravimetric analyzer (TGA, TGA/DSC3 + , Switzerland) is used to study the thermal stability of the samples at a scanning rate of 10 °C/min in the temperature range of 20–600 °C in a nitrogen atmosphere. The thermal conductivity is measured using a transient hot-wire thermal conductivity (TC3100, XIATECH, China) under 20, 25, 30, 35, 40, 45 and 50 °C. Samples are prepared into square sheets with a length of 40 mm, a width of 30 mm, and a thickness of 2 mm. Paraffin wax is detected in a beaker, and the hot wire probe is completely wrapped during melting and solidification. No data were excluded from the analyses. The tensile and compressive properties of the F-FSPCMs are measured with an electronic universal material testing machine (Instron 5985 high force universal testing machine, USA). Each tensile sample is prepared from a mold that is 90 mm in length, 8 mm in width, and 2.5 mm thick. These samples are stretched at a 10 mm/min rate under different environmental temperatures of 14, 30, 38, and 46 °C. Further, F-FSPCMs samples are cubic blocks measuring 10 mm × 10 mm × 10 mm for the compression tests. They are compressed at a 2 mm/min rate at ambient temperatures of 26 °C and 46 °C, respectively. During 200 cyclic compression tests, the rate is adjusted to 15 mm/min. The obtained measurement data represent the average of three well runs. No statistical method was used to predetermine sample size. The hydrophobicity is measured by applying a drop sharp analysis (DSA, DSA100 KRUSS, Germany). In practice, a droplet of deionized water (2 µL) is dropped onto the surface of the tested sample at ambient temperature and repeated drop tests of each sample should be performed at least three times, then the average is calculated and error bars are also used to ensure the experiment's reliability. In the leakage test, S0 and S2 are soaked in a constant temperature water bath at 50 °C for 24 h and then dried in an electrothermal blast oven at 50 °C for 24 h, after which the weight is recorded. These steps are carried out four times, consecutively for a total of 192 h in a combined water-air environment at 50 °C.

### Experiments of F-FSPCMs modules

F-FSPCMs (S2) are cast in the mold with a polyimide (PI) heater band (with an electrothermal conversion efficiency of 95%) in the molten state, and the PI heater band is completely wrapped in the middle of the F-FSPCMs. After cooling and demolding, the two specimens are wrapped entirely, and the contact surface is tightly contained without an air layer. Then, the F-FSPCMs module is obtained. Attributed to this planar heating mechanism, the heat source, generated by the conversion of electrical energy, is fully enveloped by the F-FSPCMs, ensuring that the thermal energy produced can be greatly absorbed by this material. Note that the PI heater band in F-FSPCMs module may be inclined due to the influence of the laboratory casting procedure, resulting in thicker material on one side and thinner material on the other side before and after the same heating point. The fixed heating voltage is 12 V, the power is 22 W, the dimensions are length 70 mm and width 160 mm, and control of three different heights (3, 5, and

7 mm) module are tested. A total of 8 temperature measurement points, A1-A4 and B1-B4, are set on the front and rear surfaces of the F-FSPCMs module. To accurately monitor the input power, a high-precision DC power supply is selected. T-type Patch thermocouples are selected to measure the surface temperature of the modules. The module is placed upright on the table during the test without additional insulation. A data acquisition instrument (Keysight 34972 A, USA) connected to a computer is used for real-time temperature monitoring.

## Reporting summary

Further information on research design is available in the Nature Portfolio Reporting Summary linked to this article.

## Data availability

All the data needed to evaluate the conclusions in the paper are present in the paper and/or the Supplementary information. Source data are provided with this paper.

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

## Acknowledgements

This work was supported by the National Natural Science Foundation of China (No.52038009, Y.Y., No.52293412, T.L., No.52108097, X.C.), General Project of China Postdoctoral Science Foundation under the contract (No. 2022M722061, Z.Z.), and Chongqing Science and Technology Project (No. cstc2020jcyj-msxmX0870, Q.S.). We would like to thank Dr. Daibing Luo from the Analytical and Testing Center of Sichuan University for mechanical test. We also would like to thank Analysis and Testing Center of Southwest Jiaotong University for the Raman spectroscopy test.

## Author contributions

Y.J. and Z.Z. conceived the work and wrote the original draft. Y.J. prepared samples and conducted characterization. X.C. and Q.S. prepared experiments for the mass production of sample. Y.Y. and T.L. supervised the work and revised the manuscript. All authors contributed to the discussion of the results and writing of the manuscript.

## Competing interests

The authors declare no competing interests.

## Ethical statement

Our study involved a single participant who underwent photo sessions and temperature tests related to knee thermotherapy at the Mechanical Hall Laboratory, Southwest Jiaotong University, Chengdu, China. The randomly selected participant is a 28-year-old male. We have obtained explicit consent to publish personally identifiable information. The entire testing process lasted for 3 h, and the participant received compensation of 300 RMB for lost wages and transportation expenses.
