## [Peer Review File · Nature Communications]

Ultraflexible, cost-effective and scalable polymer-based phase change composites via chemical cross-linking for wearable thermal managementREVIEWER COMMENTS

Reviewer #1 (Remarks to the Author):

This manuscript reports a general strategy to fabricate scalable polymer-based phase change composites (PW@OBC-SEBS) by melt blending and homogeneous cross-linked methods for wearable thermal management. However, the idea of preparing PW@OBC-SEBS has already been reported (J. Energy Storage 2023, 63, 107043; Macromol. Rapid Commun., 2016, 37, 1262-1267). Although this work further chemically cross-links OBC and SEBS to construct the network, which is slightly different from these papers (J. Energy Storage 2023, 63, 107043; Macromol. Rapid Commun., 2016, 37, 1262-1267), the test results only showed a slight improvement in performance or even a decline in some property, such as phase change enthalpy, entrapment efficiency, tensile strength and modulus. Most importantly, the flexibility of PW@OBC-SEBS is only above the phase transition temperature, while showing rigidity at room temperature. To these things all, I am afraid to say the level of this manuscript is hard to match the high standard of Nature Communications. Furthermore, the following specific points need to be addressed:

1. In the legend of Fig. 2, the format is not uniform. For example, "f)" should be "f".
2. Fig. 3e exhibits that the thermal conductivity of pure PW is as high as about $0.9 \text{ W m}^{-1} \text{ K}^{-1}$ at 30°C , which far exceeds the value ($0.2\text{-}0.4 \text{ W m}^{-1} \text{ K}^{-1}$) reported in the current literature. The authors are suggested to explain this phenomenon further detail.
3. In Fig. 4b, there are many curves at each temperature. Are they repeated tests of the same sample, or test results of different samples? Authors are advised to label them clearly.
4. Strain at break rather than flexibility is more suitable as the coordinate label for Fig. 4f. Because flexibility is not only shown by strain at break.
5. The unit of enthalpy should be J/g instead of W/g in the Fig. 4 b. Please try to avoid these small but serious mistakes.
6. In Fig. 4d, some data have error bars and some do not. This is inappropriate in the same graph.
7. Some experimental conditions need to be further optimized by the authors. It can be seen from Fig. 4 that the modulus is the highest when OBS-SEBS ratio is 5:1, then the ratio transitions directly to 1:0. Why not try a higher ratio, like 10:1? Maybe the performance is better than 5:1 and 1:0.
8. There are some experimental results that the authors need to further explore the reasons in detail and explain them in the article. For example, with the contents of peroxide increased, strain at break firstly increased and then decreased. Authors should not just state the results (Line 306) and a simple explanation (Line 337), preferably supported by relevant references.
9. In general, chemical crosslinking results in increased tensile strength of the polymer. However, in this study, the peak stress decreased after reaction. Please investigate further the reason.

10. There are some format errors in the reference, such as the capital form of first letter of the title. Please revise it carefully.

Reviewer #2 (Remarks to the Author):

The authors fabricated polymer-based phase change composites with high flexibility, low cost, and potential for mass production. Some comments are as follows:

1. A nomenclature should be added to explain expressions such as T_m , T_f , T_{mp} , T_{fp} , etc.
2. Fig. 3b suggests T_m is a specific value, how did the authors get this value? The melting is a gradual process, the melting point should be a range, rather than a specific value. The case is the same for T_f .
3. Fig. 5c: the authors tested the holding temperature of modules with different thicknesses. Why did the authors choose the temperature ranges of 34-36 °C and 39-42 °C?
4. Fig. 5e: the discharging is understandable, but how is the charging achieved? Using power? That information should be supplied.
5. If the module is charged by power, what is its advantage compared with using power-heat directly? The efficiency of the module seems to be lower.
6. Fig. S8b: The authors tested the leakage of F-FSPCM, however, the results are not very clear. My suggestion is to put a piece of paper under F-FSPCM, then heat it and observe if PCM leaks to paper. These results would be clearer than the current ones.

Responses to the Referees

We greatly appreciate the referees of his/her reviews and comments. Below are the point-to-point answers to the questions and suggestions raised by the referees. In this response to the referees' comments:

1. Black color - comments from the referees
2. Blue color - responses to the comments
3. Red color - added/changed to the Manuscript and Supplementary Information

Responses to Referee #1

This manuscript reports a general strategy to fabricate scalable polymer-based phase change composites (PW@OBC-SEBS) by melt blending and homogeneous cross-linked methods for wearable thermal management. However, the idea of preparing PW@OBC-SEBS has already been reported (*J. Energy Storage* 2023, 63, 107043; *Macromol. Rapid Commun.*, 2016, 37, 1262-1267). Although this work further chemically cross-links OBC and SEBS to construct the network and different from the above-mentioned two papers, the test results showed a slight improvement in performance or even a decline in some property, such as phase change enthalpy, entrapment efficiency, tensile strength and modulus. Most importantly, the flexibility of PW@OBC-SEBS is only above the phase transition temperature, while showing rigidity at room temperature.

Response: Thank you very much for your comments and helpful suggestions.

We understand that your main concern is the novelty of our manuscript when compared to the two referenced papers that you mentioned (*J. Energy Storage* 2023, 63, 107043; *Macromol. Rapid Commun.*, 2016, 37, 1262-1267). Herein, it should be noted that the first referenced paper (*J. Energy Storage* 2023, 63, 107043) was written by some authors of our new work (*Yanping Yuan et al.*). According to your specific questions and comments, we have performed additional experiments and supplemented more convincing data to highlight the novelty and further improve the quality of our manuscript

in the revised version. We made substantial progress in method, mechanism, and performance:

1) Method innovation: We develop a novel cost-effective chemical cross-linking method to synthesize leakage-proof, ultraflexible, and thermal-insulating polymer-based phase change composites. The reviewer comments that the idea of preparing PW@OBC-SEBS has been reported in the two referenced papers (*J. Energy Storage* 2023, 63, 107043; *Macromol. Rapid Commun.*, 2016, 37, 1262-1267), but he/she also already discovered the difference between our new work and the two referenced papers, and thus gave a positive comment on our work: "...this work further chemically cross-links OBC and SEBS to construct the network and different from the above-mentioned two papers". We want to emphasize that our chemical cross-linking method has distinct difference and substantial progress when compared to these physical blending methods in the two referenced papers.

In this work, we make substantial improvements based on the previous works and has obvious novelty. The physical blending method (*Macromol. Rapid Commun.*, 2016, 37, 1262-1267) needs heavy-duty mixing machinery and has the disadvantages of non-uniform mixture and high production cost. Moreover, it is very difficult to assure the thermal and mechanical stabilities of phase change composites (i.e., liquid leakage, performance degradation, etc.) when the physical blending method is used for large-scale preparation. In contrast, our method only needs simple equipment to realize the uniform mixture, and can assure the excellent thermal and mechanical stabilities of phase change composites via chemical cross-linking.

In comparison with our previous paper (*J. Energy Storage* 2023, 63, 107043), we further explore novel chemical cross-linking effect induced by free radicals from peroxide decomposition in alkanes, and reveal the comprehensive benefits in terms of equipment, cost, and overall performance. To show the distinct difference of our method, we improve the illustration of procedure for fabricating phase change composites (**Fig. R1**) and provide the more details in Supplementary Information.

Fig. R1 Illustration of the procedure for fabricating PW@OBC-SEBS composites (F-FSPCMs).

2) Mechanism innovation: We offer new insights into the understanding of the mechanism of chemical cross-linking effect induced by free radicals from peroxide decomposition in alkanes. To our knowledge, this is the first-time fabricating F-FSPCMs with cross-linking agent to construct reinforced dual 3D network structure. We systematically investigate the cross-linking toughening mechanism of polymers under different proportions of OBC/SEBS and (or) peroxide. The involved physical blending method (as described in *J. Energy Storage* 2023, 63, 107043) is just introduced as a contrast control for our experiments to show the advantage of our new chemical cross-linking method. To justify the mechanism, we made supplementary experiments and confirm the existence of peroxide-induced chemical reactions in the revised manuscript from the following aspects:

(i) Identifying the generation of bubbles during the preparation of F-FSPCMs by theoretically and experimentally methods. **Fig. R2** shows the decomposition mechanism of Di(tert-butylperoxyisopropyl)benzene (i.e., the organic peroxide cross-linking agent

applied). It can decompose into volatile organic compounds and free radicals above 156 °C by interacting with certain polymers (RH: macromolecular chains denoting H-atoms shown in Fig. R2). Based on this characteristic temperature and the analysis of the hydrogen abstraction reaction ability of generated radicals relative to the polymers with different hydrogen donor structures, we can determine the feasibility of versatile phase change composite and experiment scheme accordingly.

Fig. R2 Decomposition mechanism of Di(tert-butylperoxyisopropyl)benzene¹.

Table R1 Decomposition properties and decomposition products of Di(tert-butylperoxyisopropyl)benzene¹.

Half-life temperatures for $t_{1/2}$ ^a (°C)	Decomposition products	Relative amount (mol/mol peroxide)	Boiling point (°C)
	Tert-butanol	1.83	84.6
156 °C for 0.1 h	Acetone	0.13	56.5
134 °C for 1 h	Methanes	1.10	-161.5
114 °C for 10 h	Di-(hydroxyl-i-propyl) benzene	0.30	—
	Acetyl hydroxy-i-propyl benzene	0.54	—
	Diacetyl benzene	0.14	120

^a Data of half-life provided by AkzoNobel Polymer Chemistry.

Table R1 shows the certain proportion of peroxide's decomposition products, where the gaseous methane and acetone are generated from the chemical reaction. Correspondingly, we further performed control-experiments of traditional physical blending method and our novel chemical cross-linking method, and found a large number of bubbles are released in the latter cases (**Fig. R3**), which is consistent with the analyzed decomposition products.

Fig. R3 Experimental phenomena in the material reaction process of F-FSPCMs at 180 °C. **a** PW-SEBS gel. **b** Further addition of OBC. **c1**, **c2**, and **c3** Peroxide at different mass contents of 0.67 wt%, 1.33 wt%, 2.0 wt% respectively in the progress of the cross-linking reaction. **d** PW@OBC-SEBS after cross-linking reaction. **e** Pure PW with 0.67 wt% peroxide. **f** PW-OBC with 0.67 wt% peroxide.

(ii) Detecting the bubble compositions and the extent of chemical cross-linking using supplemented gas chromatography (GC), gas chromatography-mass spectrometry (GC-MS), and Raman spectroscopy. To validate the composition of the bubbles produced during the experiment, we further collect gas samples near the mouth of the container using a gas bag during the preparation process.

The collected gases are then analyzed for methane and acetone content using GC and GC-MS. As shown in **Fig. R4a** and **R4b**, the levels of methane and acetone associated with the chemically crosslinked PW@OBC-SEBS are detected at 54.2 mg/m³ and 2096.0 µg/m³, respectively. This indicates an approximately 30-fold increase compared to the levels shown in physical blended PW-OBC-SEBS with almost no bubble formation,

thereby aligning with the gas species predicted in the aforesaid analysis (**Fig. R2** and **R3**).

Besides, we provide the detailed GC-MS testing data in the revised version to support this point. **Fig. R4(c-e)** show the changes in the elementary chain segment involved in different processing methods using Raman spectroscopy, and determine the degree of PW@OBC-SEBS cross-linking versus peroxide content. With increasing the peroxide content from 0 to 2.0 wt%, the peak intensities of F-FSPCMs with a fixed OBC/SEBS ratio double in the bands 1295 and 1132 cm^{-1} , indicating the chemical cross-linking effect induced by free radicals from peroxide decomposition in alkanes.

Fig. R4. Demonstration of F-FSPCMs cross-linking response. **a** GC response curve for methane detection. **b** Total ion chromatogram of acetone detected by GC-MS. **c** Raman spectra of OBC, SEBS, PW, and all fabricated samples. **d** Evolution of the relative Raman intensities for F-FSPCMs with different OBC/SEBS ratios. **e** Evolution of the relative Raman intensities for sample S2 with increasing content of peroxides.

(iii) A more comprehensive examination of mechanical properties on F-FSPCMs involving tensile strength and modulus, supplemented compression strength and modulus. **Fig. R5(a-c)** shows that the peak stress of F-FSPCMs decreases while the strain increases sharply with increasing the temperature, proving the unique thermal-induced flexibility of F-FSPCMs. We observed an interesting phenomenon that the strain at break does not show a monotonic increasing trend with the increase of peroxide content, which is due to the influence of cross-linking density on the structural performance of OBC-SEBS chain

network inside F-FSPCMs. This is also the reason for the decrease in peak stress and modulus as mentioned by the reviewer, but lower modulus is supposed to be an indicative of flexibility in our F-FSPCMs according to the detailed analysis of *comment 8* and *9*. Together with the compression experimental results in **Fig. R5(f-h)**, the configuration of OBC-SEBS and (or) peroxide ratio should present an optimal threshold for chemical cross-linking toughening (i.e., OBC-SEBS=3:1 optimal).

Fig. R5 Mechanical Properties of PW@OBC-SEBS composites (F-FSPCMs). **a** Tensile stress-strain curves of F-FSPCMs at different temperatures. **b** Influence of OBC-SEBS ratio and peroxide concentration on strain at break. **c** Effect of OBC-SEBS ratio and peroxide concentration on tensile peak stress. **d** Impact of OBC-SEBS ratio and peroxide concentration on tensile modulus. **e** Compression stress-strain curves of F-FSPCMs below the melting point. **f** Compression stress-strain curves of F-FSPCMs above the melting point. **g** Cyclic compression performance of F-FSPCMs (S2) at 46 °C compressed to 30% strain over 200 cycles.

From our supplemented experiments (**Fig. R6**), it is clearly that the overall mechanical properties of F-FSPCMs via chemical cross-linking exhibit enhancement compared to that of physical blending, contributing to strengthening the understanding of chemical cross-linking toughening mechanism.

Fig. R6 Tensile test results of physical blended PW-OBC and cross-linked PW@OBC at 14 °C, 30 °C, 38 °C, and 46 °C. **a** Comparison of strain at break. **b** Comparison of peak stress. **c** Comparison of modulus.

Furthermore, we want to emphasize that the thermoplastic elastomer cross-linking induced by radical generation from peroxide decomposition in alkanes is a versatile strategy for fabricating F-FSPCMs. The analogous organic compounds including styrene-butadiene-styrene (SBS), polyolefin elastomer (POE), and even cheaper plastic, e.g., high-density polyethylene (HDPE) and low-density polyethylene (LDPE) can also be implemented to synthesize phase change composites.

Besides the SEBS/OBC raw materials, we further perform experiments to validate the feasibility of our thermoplastic elastomer cross-linking for synthesizing phase change composites using relatively incompatible mixtures (**Fig. R7**). The control-experiments of physical blending and chemical cross-linking techniques using PW-OBC and PW-POE-LDPE confirm the distinct advantages of reproducibility, efficient preparation processes, and broad raw material choices in our chemical cross-linking method.

Fig. R7 Tensile test results of physical blended PW-POE-LDPE and cross-linked PW@POE-LDPE at 27 °C, 37 °C and 47 °C. **a** Comparison of strain at break. **b** Comparison of peak stress. **c** Comparison of modulus.

3) Superior performance: Base on the innovations in both method and mechanism of our chemical cross-linking strategy, we realize a distinct enhancement in the flexibility of F-FSPCMs within a wide range of temperature below and above the melting point temperatures, superior to the two referenced papers (*J. Energy Storage* 2023, 63, 107043; *Macromol. Rapid Commun.*, 2016, 37, 1262-1267). More importantly, our phase change composites show high energy density, and the enthalpy in our work (175 ± 5 J/g) is approximately 1.3-2.0 times higher than the reported values in the two referenced papers (~ 140 J/g and ~ 80 J/g).

Furthermore, it is worth noting that the flexibility of phase change composites near the phase transition temperature plays a crucial role in thermal management for practical applications. The strain at break of our F-FSPCMs near phase transition temperature is as high as 374.73% at 46 °C, approximately 15 times higher than the strain at break below melting point (23.24% at 14 °C). Although the strain at break of our F-FSPCMs below melting point is 23.24%, we want to emphasize that this figure indicates a remarkable improvement of 46% when compared to the values obtained from these physical blending methods (*Macromol. Rapid Commun.*, 2016, 37, 1262-1267; *J. Energy Storage* 2023, 63, 107043). Besides, this value would increase exponentially to a higher level as

the temperature increases to near the phase transition temperature.

Moreover, we found and identified that PW@POE-LDPE formulations exhibit enhanced flexibility below its phase transition temperature (i.e., data at 37 °C) during our exploration of alternative organic fillers, as shown in **Fig. R7a and R7b**. Such interesting characteristic is indeed worthy to be in-depth investigated in our research works in the near future. Much thanks for your suggestions!

In a word, our work makes substantial progress in concept, new method, fundamental mechanism, and superior performance. The proposed novel chemical cross-linking method provides new insights for designing flexible phase change composites and exhibits great potential in large-scale applications with low cost and high reliability. To further emphasize the distinctive feature of chemical cross-linking in our work, the term "chemical cross-linking" is incorporated into the manuscript title: The initial title of "*Ultraflexible, cost-effective, and scalable polymer-based phase change composites for wearable thermal management*" has been changed to: "*Ultraflexible, cost-effective, and scalable polymer-based phase change composites via chemical cross-linking for wearable thermal management*". We have also improved the language and some expressions of the manuscript.

Changes add to manuscript:

Fig. R4a is added in the revised manuscript as **Fig. 2a**

Fig. R4b is added in the revised manuscript as **Fig. 2b**

References 2-4 are added in the revised manuscript as **References 32-34**

Recently, the preparation of functional materials (e.g., elastic ferroelectric materials) via cross-linking toughening to meet the performance enhancement for flexible wearable devices has gradually attracted attention, and such effect induced by C-C bond generation between polymers is worth exploring and utilizing in aforesaid polymer-based phase change material³²⁻³⁴. **(Page 6 Line 118)**

To validate the composition of gas bubbles produced during the experiment, we collect gas samples near the mouth of the container using a gas bag during the preparation

process. The collected gases are then analyzed for methane and acetone content using gas chromatography (GC) and gas chromatography-mass spectrometry (GC-MS). As depicted in **Fig. 2a** and **2b**, the levels of methane and acetone associated with the chemically crosslinked PW@OBC-SEBS are detected at 54.2 mg/m³ and 2096.0 µg/m³, respectively. This represents an approximately 30-fold increase compared to the levels found in physical blended PW-OBC-SEBS. It is noteworthy that peroxides only generate decomposition products upon abstracting hydrogen atoms from the polymer, thereby aligning with the gas species predicted in the theoretical analysis (seen in Supplementary **Fig. S1** and **Table S1**). (Page 10 Line 181).

Methane production is quantified using a gas chromatograph (GC, SP3420A, bfrl, China) equipped with a Flame Ionization Detector (FID). The chromatographic conditions employ a GDX-104 column, with a column temperature of 50 °C, an injector temperature of 100 °C, and a detector temperature of 200 °C. The split ratio is set at 10:1, and the injection volume is 1.0 mL. Acetone concentration is measured using a gas chromatography-mass spectrometry system (GC-MS, 8890-5977B, Agilent, USA). A 400 mL sample is introduced into a cryogenic trap concentrator, and 50 mL of a standard gas for internal standard method is simultaneously injected. The conditions are set with a DB-1 column, an injector temperature of 150 °C, a column flow rate of 1 mL/min, and a split ratio of 20:1. The temperature program is initiated at 5 °C, held for 15 min, ramped at 8 °C/min to 180 °C, further increased at 10 °C/min to 220 °C, and then held for 9 min. (Page 31 Line 576).

Changes add to Supplementary Information (SI):

Fig. R5 is added in SI as Supplementary **Fig. S12**

Fig. R6 is added in SI as Supplementary **Fig. S13**

During our comparative validation experiments, we further employ OBC (20 wt%) and PW (80 wt%) as raw materials, and subject them to both physical blending and chemical crosslinking processes (with peroxide proportion of 0.67 wt%), resulting in two distinct composites, i.e., PW-OBC and PW@OBC, respectively. From the tensile

performance results corresponding to four characteristic temperature points (as depicted in **Fig. S12**), the chemical cross-linked PW@OBC consistently exhibits higher strain at break compared to its physical mixed counterpart. Conversely, there is a marked decrease in its peak stress and modulus. (**Page 18 Line 219**)

Meanwhile, the universality tests involving the usage of polyolefin elastomer (POE, ExxonMobil 6202, 12.5 wt%), low-density polyethylene (LDPE, LG MB-9500, 12.5 wt%), and PW (75 wt%) as raw materials have also undergone both physical blending and chemical crosslinking treatment, then resulting in two distinct PW-POE-LDPE and PW@POE-LDPE composites, respectively. As illustrated in **Fig. S13**, a similar pattern emerges, that is, cross-linked specimens exhibit greater strain at break compared to their physical mixed counterparts, while peak stress and modulus display a marginal reduction. However, it is noteworthy that the strain at break rate does not follow a continuous increase with temperature. In the case of PW@POE-LDPE, the strain at break rate is higher at 37 °C compared to that of 47 °C, a performance characteristic that is not initially predicted. (**Page 19 Line 226**)

The above results illustrate that the peroxide chemical cross-linking method can render the material "more flexible" by improving strain at break while reducing elastic modulus and peak stress. This enlightens the great potential of chemical cross-linking in transcending the constraints of physical blending. Not only one or two, but even three or more thermoplastic elastomers can derive synergistic effects. That is to say, engaging other classes of thermoplastic elastomers may lead to more unpredictable performance, which is indeed worthy of in-depth exploration. (**Page 19 Line 235**)

Detailed remarks

Comment 1: In the legend of Fig. 2, the format is not uniform. For example, "f)" should be "f".

Response: Thank you for your kind reminding. The typo of **Fig. 2** has been modified and its title has been updated as follows. We also carefully go through the manuscript to avoid

the similar situations.

Changes add to manuscript:

Fig. 2 Demonstration of F-FSPCMs cross-linking response. **a** GC response curve for methane detection. **b** Total ion chromatogram of acetone detected by GC-MS. **c** Raman spectra of OBC, SEBS, PW, and all fabricated samples. **d** Evolution of the relative Raman intensities for F-FSPCMs with different OBC/SEBS ratios. **e** Evolution of the relative Raman intensities for sample S2 with increasing content of peroxides. **(Page 13 Line 218)**

Comment 2: Fig. 3e exhibits that the thermal conductivity of pure PW is as high as about $0.9 \text{ W m}^{-1} \text{ K}^{-1}$ at $30 \text{ }^\circ\text{C}$, which far exceeds the value ($0.2\text{-}0.4 \text{ W m}^{-1} \text{ K}^{-1}$) reported in the current literature. The authors are suggested to explain this phenomenon further detail.

Response: Thank you for your comprehensive review. As indicated by the reviewer, we re-evaluate the thermal conductivity of paraffin wax (PW), and the obtained value at $30 \text{ }^\circ\text{C}$ is in the range of $0.3\text{-}0.4 \text{ W m}^{-1} \text{ K}^{-1}$.

We contacted engineers from NETZSCH and were advised to use transient hot-wire method to measure the thermal conductivity of PW. The laser flash and transient hot-wire methods provide different ranges and applicability in measuring thermal conductivity. Where the laser flash method is commonly employed for materials with relatively high thermal conductivities ($>10 \text{ W m}^{-1} \text{ K}^{-1}$). However, this method tends to yield higher measurement errors when measuring materials with low thermal diffusivity. Moreover, the laser flash method is generally suitable for measuring the thermal conductivity of isotropic materials and cannot be used for anisotropic materials. In contrast, the transient hot-wire method is well-suited for measuring the thermal conductivities of solid and liquid materials with low thermal conductivities.

We employed transient hot-wire thermal conductivity apparatus to measure the thermal conductivities of paraffin wax and PW@OBC-SEBS at a temperature range from $20 \text{ }^\circ\text{C}$ to $50 \text{ }^\circ\text{C}$ (**Fig. R8**). As suggested, the original **Fig. 3e** and the relevant statements have been modified in the revised manuscript.

Fig. R8 Thermal conductivity and diffusivity comparison of PW and PW@OBC-SEBS (S2).

Changes add to manuscript:

Fig. R8 is added in the revised manuscript as **Fig. 3e**

Furthermore, incorporating a dual three-dimensional cross-linking network of OBC-SEBS into the PCM still maintains a relatively low thermal conductivity for F-FSPCMs (seen in **Fig. 3e**). The thermal conductivity of F-FSPCMs remains within the range of 0.23-0.27 W m⁻¹ K⁻¹ before the phase transition (at 35-40 °C), and experiences a slight decrease to 0.16 W m⁻¹ K⁻¹ after the phase transition (> 45 °C), thus contributing to the effective thermal insulation for personal thermal management. (**Page 17 Line 316**)

The thermal conductivity is measured using a transient hot-wire thermal conductivity (TC3100, XIATECH, China) under 20, 25, 30, 35, 40, 45 and 50 °C. F-FSPCMs are prepared into square sheets with a length of 40 mm, a width of 30 mm, and a thickness of 2 mm. Paraffin wax is detected in a beaker, and the hot wire probe is completely wrapped during melting and solidification. (**Page 32 Line 612**)

Comment 3: In Fig. 4b, there are many curves at each temperature. Are they repeated tests of the same sample, or test results of different samples? Authors are advised to label them clearly.

Response: Thank you for the comment and suggestion. The presented test results describe

the curves of samples with different OBC/SEBS ratios under identical chemical cross-linking conditions (denoted as S0-S3). Considering the presence of a certain number of 16 curves and their dense parts, distinct background shadows are employed to distinguish four temperature regions: 14, 30, 38, and 46 °C, as depicted in **Fig. R9**. Different line colors indicate the variations in sample characteristics at same temperatures.

Fig. R9 Trendline clusters of stress-strain in variable temperature tensile testing.

Changes add to manuscript:

Fig. R9 is added in the revised manuscript as **Fig. 5a**

Comment 4: Strain at break rather than flexibility is more suitable as the coordinate label for Fig. 4f. Because flexibility is not only shown by strain at break.

Response: Many thanks for your kind comments. As the reviewer suggested, "Flexibility" is a broadly descriptive term that can be characterized using various parameters. In the context of flexible form-stable phase change materials (F-FSPCMs), it can be manifested through bending, twisting, stretching, and compression. However, these properties are often illustrated solely through images in literatures. For quantitative comparison, many materials face limitations due to sample preparation and testing constraints, making it challenging to conduct extensive measurements. On the other hand, "strain at break" is a common method to quantitatively represent flexibility, and thus we previously used "strain at break" to characterize and compare the "flexibility" effects similar to that in literatures.

Undoubtedly, your feedback has prompted us to re-evaluate the material compositions and testing protocols concerning flexibility representation. Firstly, we have modified the X-axis label of the original **Fig. 4f** (as depicted in **Fig. R10**) to make the description more concrete. Secondly, we further expand the mechanical compression performance testing for F-FSPCMs at different temperatures to provide more comprehensive data, which is expected to highlight the "Ultra-flexibility" feature dominated by the chemical cross-linking toughening mechanism (seen in **Fig. R11**). Specially, we measured the compression characteristics at 26 °C and 46 °C, presenting relatively stable cyclic compression performance over 200 cycles at 46 °C. This fatigue testing is a valuable indication of stability for subsequent thermotherapy applications. References 21-24 have been supplemented here to support the discussion of **Fig. R11**.

Fig. R10 Comparison of both flexibility and enthalpy of our F-FSPCMs with those in literature⁵⁻²⁰.

Fig. R11 Mechanical Properties of PW@OBC-SEBS composites (F-FSPCMs). **a** Tensile stress-strain curves of F-FSPCMs at different temperatures. **b** Influence of OBC-SEBS ratio and peroxide concentration on strain at break. **c** Effect of OBC-SEBS ratio and peroxide concentration on tensile peak stress. **d** Impact of OBC-SEBS ratio and peroxide concentration on tensile modulus. **e** Comparison of strain at break and enthalpy of our prepared F-FSPCMs with literature⁵⁻²⁰. And the detailed information is shown in **Table S3** (Supplementary). **f** Compression stress-strain curves of F-FSPCMs below the melting point. **g** Compression stress-strain curves of F-FSPCMs above the melting point. **h** Influence of OBC-SEBS ratio and peroxide concentration on compression peak stress. **i** Effect of OBC-SEBS ratio and peroxide concentration on compression modulus. **j** Cyclic compression performance of F-FSPCMs (S2-0.67) at 46 °C compressed to 30% strain over 200 cycles.

Changes add to manuscript:

Fig. R10 is added in the revised manuscript as **Fig. 5e**

Fig. R11f is added in the revised manuscript as **Fig. 5f**

Fig. R11g is added in the revised manuscript as **Fig. 5g**

Fig. R11j is added in the revised manuscript as **Fig. 5h**

References 22-24 are added in the revised manuscript as **References 55-57**

To further validate the mechanical characteristics of F-FSPCMs, mechanical tensile and compressive assessments are performed on F-FSPCM specimens at different temperatures. **Fig. 5a** depicts stress-strain profiles and schematic illustrations elucidating the tensile testing procedure. The results clearly indicate that the F-FSPCMs exhibit reduced peak stress and modulus as well as increased strain with increasing temperature (**Fig. 5b-5d**). Notably, significant differences in strain at break are observed for S0-S3 at the same temperatures. Specifically, SEBS content can enhance ductility significantly, the strain at break can be increased from 75% to 560% at 46 °C. On the other hand, as the peroxide content increases from 0 wt% to 2 wt%, there is an initial increase followed by a decrease in strain at break, which was consistently observed across all temperatures due to the heightened cross-linking density. A low cross-linking density positively impacts network structure performance, while excessive cross-linking density would in turn restrict molecular chain slippage, consequently limiting the mechanical properties of the blend. This observation aligns with established principles in previous researches, indicating the existence of an optimal degree of mild cross-linking that can be used to regulate fracture elongation to achieve desired values^{33,42}. (**Page 19 Line 341**)

Fig. 5e presents a comprehensive comparison of the maximum latent heat and degree of flexibility achievable by various typical shaped matrixes^[24-26, 41, 43-54]. Encouragingly, our F-FSPCMs exhibit higher enthalpies and stretchability, and have breaking strain tolerance of 15-133% without fracture or damage below the phase change temperature, while the breaking strain can also further increase from 69% to 156% or 144% to 560% near or above the phase change temperature, respectively. (**Page 22 Line 395**)

The F-FSPCMs exhibit distinct mechanical behaviors before and after the phase change under compression, as depicted in **Fig. 5f** and **Fig. 5g**. At a temperature below phase change point, increasing SEBS content can improve the compressive strength

continuous deformation of F-FSPCMs during the compression process, eventually resulting in a flattened rectangular prism shape. However, at a temperature above phase change point, the F-FSPCMs manifest internal cracks and defects as the compressive strain approaches nearly 20% when the OBC/SEBS ratio surpasses 3:1, leading to a decrease in peak compressive stress. Conversely, at ratios of 3:1 and 1:1, the F-FSPCMs exhibit similar elastomeric characteristics with superior compressive strength. Moreover, after undergoing 200 cycles of 30% compression fatigue testing, the F-FSPCMs consistently followed the same stress-strain curve during both loading and unloading processes (**Fig. 5h**), making them a promising choice for thermotherapy module applications that demonstrate substantial deformability and high stability under repetitive compression. (**Page 23 Line 407**)

Notably, as the degree of chemical cross-linking increases below the phase transition temperature, both compressive and tensile modes gradually decrease peak stress and elastic modulus. Conversely, there are no significant changes above the phase transition temperature (46 °C) (see Supplementary **Fig. S11**). The higher tensile stress and modulus are primarily related to the degree of entanglement of central chain segments within the elastomer⁵⁵. During the cross-linking process, OBC-SEBS molecular segments undergo dynamic fracture and recombination, forming new C-C single bonds and reducing the polymer's average molecular weight, leading to a relative decrease in stress intensity and modulus. As the degree of cross-linking increases, cross-linking reactions become more pronounced and molecular segments tend to experience more significant disruption, ultimately reducing their mechanical properties. Therefore, it should be emphasized that the regulatory mechanism of "chemical cross-linking" does not follow a fixed pattern, especially considering that the paraffin content in our phase change composites accounts for 80% rather than simply a single thermoplastic elastomer cross-linking system.

Lower elastic modulus and higher strain at break can be regarded as indication of flexibility for our F-FSPCMs base on statements in cross-linking relevant researches^{56,57}. Further, we observed similar characteristics in other material systems (PW-OBC, PW-POE-LDPE) (Supplementary **Fig. S12** and **Fig. S13**). Specifically, chemical cross-linking driven by peroxide decomposition reaction exhibits a substantial enhancement effect on

flexibility compared to physical blending. This insight suggests that, beyond the choice of raw materials and their proportions, the utilization of peroxide cross-linking in thermoplastic elastomers or cost-effective plastics can serve as a universal means for engineering flexible phase change materials. (Page 24 Line 435)

The tensile and compressive properties of F-FSPCMs are measured with an electronic universal material testing machine (Instron 5985 high force universal testing machine, USA). Each tensile sample is prepared from a mold that is 90 mm in length, 8 mm in width, and 2.5 mm thick. These samples are stretched at a 10 mm/min rate under different environmental temperatures of 14, 30, 38, and 46 °C. Further, F-FSPCMs samples are cubic blocks measuring 10 mm × 10 mm × 10 mm for the compression tests. They are compressed at a 2 mm/min rate at ambient temperatures of 26 °C and 46 °C, respectively. During 200 cyclic compression tests, the rate is adjusted to 15 mm/min. The obtained measurement data represent the average of three well runs. (Page 33 Line 619)

Changes add to Supplementary Information (SI):

Fig. R11h is added in SI as Supplementary **Fig. S11a**

Fig. R11i is added in SI as Supplementary **Fig. S11b**

Comment 5: The unit of enthalpy should be J/g instead of W/g in the Fig. 4b.

Response: Thank you for your kind reminding. We have corrected this part from W/g to J/g and renumbered **Fig. 4f** as **Fig. 5e** in the revised manuscript. Also, we carefully go through the manuscript to avoid the similar situations. Kindly find the updated figure listed in **Fig. R10**.

Changes add to manuscript:

Fig. R10 is added in the revised manuscript as **Fig. 5e**

Comment 6: In Fig. 4d, some data have error bars and some do not. This is inappropriate in the same graph.

Response: Thank you for the comment. We admit that the original manuscript lacks a more solid examination for **Fig. 4d**, making the actually provided small size error bars for certain data points almost unrecognizable. Anyway, this is in turn a positive indication that measurements under those conditions are highly stable and reliable. As suggested, to improve recognition and make the results more convincing, we have incorporated breaks in the axes of figure and renumbered original **Fig. 4d** in the revised manuscript (seen in **Fig. R12**). Furthermore, we have attached the raw data, including the mean values and standard deviations from three separate measurements.

Fig. R12 Effects of OBC-SEBS ratio and peroxide concentration on peak tensile stress.

Changes add to manuscript:

Fig. R12 is added in the revised manuscript as **Fig. 5c**

Comment 7: Some experimental conditions need to be further optimized by the authors. It can be seen from Fig. 4 that the modulus is the highest when OBS-SEBS ratio is 5:1, then the ratio transitions directly to 1:0. Why not try a higher ratio, like 10:1? Maybe the performance is better than 5:1 and 1:0.

Response: Thank you for the comment. As the reviewer suggested, the authors realize that it is necessary to provide a comprehensive examination of modulus to illustrate the

mechanical property of our F-FSPCMs. It should be first emphasized that through multiple repeated measurements in the original manuscript, the machine consistently displays modulus ~ 150 MPa for the 5:1 ratio group at 30 °C after excluding test errors and fluctuations. Then following your advice, we further prepare a series of F-FSPCMs samples with ratios of 7:1 and 10:1, but even more unstable modulus data is presented at 30 °C. In this case, we revisit this issue and consult with Instron engineers, only to realize that the instability of modulus data is due to an issue with the algorithm used by the machine in the "Elastic Modulus (Auto Young's)" mode.

To be specific, the algorithm for "Auto Young's" is as follows: Firstly, it takes the first data point greater than or equal to 2% of the maximum load as the starting point and either the yield or maximum load point as the ending value. Secondly, it divides the data on the stress axis into six regions and calculates the slope for each region using the least squares method. Finally, the region with the maximum slope is determined to be the position of the modulus.

As shown in **Fig. R13**, following the above algorithm, the maximum load for sample S1 is 1.06 MPa, and the first calculation region is between 0.0212 MPa and 0.194 MPa, resulting in a calculated modulus of 156.71 MPa (Linear fit 1 in Fig. R13). This erroneous modulus arises from the stress spikes, caused by the self-locking effect of fixture and sample at the beginning of tension phase. After that, the machine automatically provides a modulus of 156.71 MPa, while the actual modulus should be the slope in the 2%-8% linear phase, which is actually ~ 7.91 MPa (Linear fit 2 in Fig. R13). Therefore, we reconfirm the objectively stable force-displacement curves for all tensile and compression samples, and import the raw data into OriginPro Software for linear fitting in the elastic stage, thereby obtaining more reasonable and accurate modulus values. The final results can be seen in **Fig. R14** as below. Also, relevant statements have been modified in the revised manuscript.

Fig. R13 The algorithm for "Elastic Modulus (Auto Young's)" mode.

Fig. R14 Effects of OBC-SEBS ratio and peroxide concentration on **a** tensile modulus and **b** compression module of PW@OBC-SEBS composites (F-FSPCMs).

Changes add to manuscript:

Fig. R14a is added in the revised manuscript as **Fig. 5d**

For your convenience and to ensure clarity, the revisions to the manuscript have been supplemented in **Comment 4**, without the need to filter through lengthy paragraphs again.

Changes add to Supplementary Information (SI):

Fig. R14b is added in SI as Supplementary **Fig. S11b**

Comment 8: There are some experimental results that the authors need to further explore the reasons in detail and explain them in the article. For example, with the contents of peroxide increased, strain at break firstly increased and then decreased. Authors should not just state the results (Line 306) and a simple explanation (Line 337), preferably supported by relevant references.

Response: Thanks for your suggestion. We observed that the strain at break initially increases and then decreases with increasing peroxide content. This non-linear behavior prompts pertinent inquiries regarding the underlying mechanisms that govern the mechanical characteristics of crosslinked F-FSPCMs.

For thermoplastic elastomers and rubber materials, it is a typical pattern that the strain at break initially rises and subsequently drops with increasing the degrees of cross-linkings^{3,21}. This trend can be attributed to the intricate balance between cross-linking density and the migration rate of OBC-SEBS chains²⁵. An initially reduced cross-linking density enhances the structural integrity of OBC-SEBS network, elevating its tensile fracture capacity. However, excessive cross-linking diminishes chain migration rates, augmenting rigidity and resulting in a reduced fracture strain.

It is noteworthy to identify optimal threshold for minimal cross-linking to adjust the fracture strain with desired levels finely. This insight contributes to a deeper understanding of the cross-linking system and underscores the need for precise control over peroxide content to achieve desired mechanical performance in practical applications.

Changes add to manuscript:

For your convenience and to ensure clarity, the revisions to the manuscript have been supplemented in **Comment 4**, without the need to filter through lengthy paragraphs again.

Comment 9: In general, chemical crosslinking results in increased tensile strength of the polymer. However, in this study, the peak stress decreased after reaction. Please investigate further the reason.

Response: Thank you for your feedback. Strength and toughness represent two trade-off properties of a material, typically exhibiting an inverse relationship². Chemical cross-linking can manifest varying mechanical properties under different polymers and types of cross-linking agents. For instance, the introduction of metal-ligand sacrificial bonds into a four-component multiblock copolymer results in enhanced tensile strength and elastic modulus, albeit with a slight reduction in strain in the elastic fibers²⁶. While vulcanized rubbers modified using a novel semisolid preparation method exhibit significantly increased tensile strength but reduced strain, accompanied by a lower Young's modulus and improved flexibility²⁴. These examples illustrate the diverse mechanical responses achievable through chemical cross-linking under distinct polymer systems and cross-linking methodologies.

In our investigation of F-FSPCMs, we observed that their strain at break exhibits significant improvement compared to physical blending. This enhancement initially increased and then decreased with increasing cross-linking degree, as explained in Question 8. Both peak stress and modulus consistently decreased at a temperature below phase transition temperature, while no significant changes were observed above this temperature (46 °C) (see **Fig. R5** and **Fig. R14**). The higher tensile stress and modulus are primarily related to the degree of entanglement of central chain segments within the elastomer²². During cross-linking, OBC-SEBS molecular segments undergo dynamic fracture and recombination, forming new C-C single bonds and reducing the polymer's average molecular weight, leading to a relative decrease in stress intensity and modulus. As the degree of cross-linking increases, molecular segments tend to experience more significant disruption, ultimately reducing their mechanical properties. Therefore, we want to emphasize that the regulatory mechanism of "chemical cross-linking" does not follow a fixed pattern, especially considering that the paraffin content in our phase change composites accounts for 80% rather than simply a single thermoplastic elastomer cross-

linking system.

In this case, lower elastic modulus and higher strain at break can be regarded as indication of flexibility. It should be noted that enhancing flexibility often comes at the expense of strength, necessitating optimizing the inherent balance within cross-linked polymer networks³. As supplementary evidence, we observed similar characteristics in other material systems (PW-OBC, PW-POE-LDPE) (see Supplementary **Fig. R6** and **Fig. R7**). Specifically, chemical cross-linking driven by peroxide decomposition reaction exhibits a significant enhancement on flexibility compared to conventional physical blending.

Changes add to manuscript:

For your convenience and to ensure clarity, the revisions to the manuscript have been supplemented in ***Comment 4***, without the need to filter through lengthy paragraphs again.

Comment 10: There are some format errors in the reference, such as the capital form of first letter of the title. Please revise it carefully.

Response:

Thank you for the kind reminding. We have corrected the format errors in the reference. We also carefully go through the manuscript and supplementary information to avoid the similar situations.

Responses to Referee #2

The authors fabricated polymer-based phase change composites with high flexibility, low cost, and potential for mass production.

Response: Thank you for your constructive feedback and recognizing the features of our polymer-based phase change composites. We deeply appreciate your time and patience in the review process. We have modified our manuscript according to your rigorous comments and suggestions. Our responses are listed below and the revised portion is marked in red in the revised manuscript. Again, thank you so much for helping us improve our manuscript.

Detailed remarks

Comment 1: A nomenclature should be added to explain expressions such as T_m , T_f , T_{mp} , T_{fp} , etc.

Response: Thank you for your kind reminding. According to your suggestion, a list of abbreviations is below. Considering that similar tables are rarely listed in the main text of Nature Communications, we made a statement in the revised manuscript about the description of all abbreviations: "The definition specification of T_m and T_f can be found in Supplementary Information (Fig. S10)" (also see Comment 2 Fig. R15). Please find on Page 15 Line 274 in the revised Supplementary Information with changes marked.

Nomenclature			
PCMs	phase change materials	T_m	melting temperature
FSPCMs	form-stable phase change materials	T_{mp}	peak melting temperature
F-FSPCMs	flexible form-stable phase change materials	T_f	freezing temperature
TPEs	thermoplastic elastomers	T_{fp}	peak freezing temperature
PW	paraffin wax	ΔH_m	melting enthalpy
OBC	olefin block copolymer	ΔH_f	freezing enthalpy
SEBS	styrene-ethylene-butylene-styrene		
POE	polyolefin elastomer		
LDPE	low-density polyethylene		

Comment 2: Fig. 3b suggests T_m is a specific value, how did the authors get this value? The melting is a gradual process, the melting point should be a range, rather than a specific value. The case is the same for T_f .

Response: Thank you for the rigorous comment. As stated by the reviewer, for phase change materials (PCMs) or their composites, the melting and freezing stages are basically gradual processes. Where the melting and freezing temperature range of PCMs can be determined using a differential scanning calorimeter (DSC) through the dynamic measurement method. In our work, the T_m , T_{mp} , and T_{m-end} represent the extrapolated onset temperature, peak temperature, and extrapolated end temperature obtained from the main endothermic peak of PCMs in DSC measurements, respectively. Similarly, for the exothermic phase transition, i.e., solidification regime, there are corresponding values of T_f , T_{fp} , and T_{f-end} .

Essentially, both T_{mp} and T_{m-end} are often influenced by the mass of test sample and the heating rate in DSC measurements. With an increase in the heating rate, the onset temperature of melting peak changes insignificantly, but the peak and end temperatures rise, resulting in a broader peak shape. In this case, “ $T_m = T_{onset}$ ” has come to be defined as the phase transition temperature in different literatures^{27,28}, while our study presents both T_m and T_{mp} values to clarify this point.

Furthermore, it should be noted that the paraffin selected here is unique because of its hybrid alkane composition characteristics, where the main peak is typically formed by the overlap of two peaks. As shown in **Figure R15** (Supplementary **Fig. S10**), two extrapolated temperatures (points A and B) and two extrapolated temperatures (points E and G) appear during the endothermic and exothermic phase transition, respectively. After that, considering the significant weightage of the single peak area at points B and E, which dominates the corresponding phase transition process and therefore, the temperature at points B and E are reasonably defined as T_m and T_f , respectively.

Fig. R15 Definition of phase transition temperature T_m , T_{mp} , T_f , and T_{fp} in DSC.

Base on your valuable comments, we supplement **Fig. R15** and the aforesaid statements to highlight the precise definition of these thermal parameters in the revised Supplementary Information. Besides, we add a sentence to emphasize this point in the revised manuscript as follows.

Changes add to manuscript:

The definition specification of T_m and T_f can be found in Supplementary Information (**Fig. S10**). (Page 15 Line 274)

Changes add to Supplementary Information (SI):

Fig. R15 is added in SI as Supplementary **Fig. S10**

Comment 3: Fig. 5c: the authors tested the holding temperature of modules with different thicknesses. Why did the authors choose the temperature ranges of 34-36 °C and 39-42 °C?

Response: Thank you for your valuable question. To this issue, it is necessary to point out that the paraffin utilized in our study is a mixture of alkanes with varied carbon atoms, which exhibits two distinct peaks during the exothermic phase transition, as depicted in the above **Fig. R15**. Specifically, the temperatures corresponding to points E and G are T_f values of 42.1 °C and 34.3 °C, respectively, reflecting inherent characteristics of paraffin.

Fig. R16 Surface temperature of a certain F-FSPCMs module during a heat storage/discharge cycle

As illustrated in **Fig. R16**, when measuring the surface temperature of the F-FSPCMs temperature control module, two temperature plateaus are observed due to the isothermal nature of the phase change process. For the demonstration of our F-FSPCMs for wearable thermal management like thermal therapy herein, the temperature range of 39-42 °C represents the desired constant temperature plateau, while the range of 34-36 °C corresponds to the concomitant constant temperature plateau, aligning with the data obtained from the DSC curves. Anyway, the results obtained from lower temperature region can also provide data support for other personal thermal management scenarios. It is important to note that the surface temperature is solely determined by the properties of our F-FSPCMs, while the module's thickness only influences the duration for which a specific phase change temperature is maintained. Based on your kind comment, we add some descriptions of the experimental basis involved in revised **Fig. 6c** to improve this point.

Changes add to manuscript:

We further analyzed the effect of module thickness on the two-stage constant surface temperature (**Fig. 6c**). The obtained results provide a guideline for the design of modular specifications to meet practice requirements. (**Page 27 Line 491**)

Comment 4: Fig. 5e: the discharging is understandable, but how is the charging achieved? Using power? That information should be supplied.

Response: Thank you for your rigorous comment and kind suggestion. We are sorry that we didn't explain these issues clearly in the original manuscript. We have revised the initial two-dimensional schematic representation for the discharging and charging (i.e., **Fig. 5a** in the original manuscript) into a three-dimensional illustration according to your advice, as depicted in **Fig. R17 (Fig. 6a** in the revised manuscript). Specially, the polyimide (PI) heating sheet (or PI heater band) is a planar flexible heating element composed of resistive material sandwiched between two layers of PI film, produced through a high-temperature and high-pressure lamination process. This heating element exhibits excellent insulation strength, electrical resistance, and resistance stability.

We further integrated the F-FSPCMs with the PI heater band in a molten state within a mold in our approach. Importantly, the PI heating element can be entirely encapsulated within the F-FSPCMs. This is because upon cooling and solidification, such two components (i.e., F-FSPCMs and PI heater band as seen in **Fig. R17**) can form a tight, air-free bond at their interface, and consequently, a flexible and contour-controlled phase change module is widely available. Subsequent experimental investigations have demonstrated a favorable compatibility between F-FSPCMs and the PI heater band.

Fig. R17 Experimental setup for the evaluation of heat storage/release of the F-FSPCMs module.

The aforementioned flexible contoured phase change temperature control module operates through a planar heating mechanism. The heat source, generated by the conversion of electrical energy, is fully enveloped by the F-FSPCMs, ensuring that the thermal energy produced can be greatly absorbed by this material. In our thermal

management experiments, the surface of the F-FSPCMs module demonstrated the ability to regulate skin surface temperatures over extended periods.

For better illustrate the approach and the results, we reinforce our manuscript and supplement the isolated heating performance of the PI heater band (seen in **Fig. R18** as below). Upon electrification, the surface of the PI heater band can reach temperatures up to 110 °C within 5 min. However, once the power is disconnected, the surface temperature rapidly declines to below 30 °C in a span of just 3.5 min. Therefore, such comparative experiment further reflects the excellent long-term temperature control capability of the F-FSPCMs module for wearable thermal management. As the reviewer suggested, we have supplemented relevant statements in the revised manuscript to improve this point. Renumbered **Fig. 6a** and **Fig. 6e** have also been modified to enhance readability.

Fig. R18 Temperature evolution of insulating device based on F-FSPCMs module (5 mm thickness) acting on a knee during charging and discharging and the comparison with the isolated heating performance of the PI heater band.

Changes add to manuscript:

Fig. R17 is added in the revised manuscript as **Fig. 6a**

Fig. R18 is added in the revised manuscript as **Fig. 6e**

Upon cooling and solidification in our module preparation, the two components, i.e., F-FSPCMs and PI heater band, can form a tight, air-free bond at their interface, and consequently, a flexible and contour-controlled phase change module is widely available.

(Page 25 Line 466)

Attributed to this planar heating mechanism, the heat source, generated by the conversion of electrical energy, is fully enveloped by the F-FSPCMs, ensuring that the thermal energy produced can be greatly absorbed by this material. **(Page 34 Line 644)**

Comment 5: If the module is charged by power, what is its advantage compared with using power-heat directly? The efficiency of the module seems to be lower.

Response: Many thanks for your kind comments. As mentioned by the reviewer, there is no doubt that using power-heat directly can achieve the goal of charging, but it is difficult to meet the effect of long-term constant temperature control ability without the need for frequent operations. That is to say, it lacks a function to a certain extent. In contrast, the core capability of PCMs in energy storage applications is “peak shaving and valley filling”, which can reuse energy in a temporal dimension. For instance, in the case of high-density heat flux, heating PCMs and insulating them can release a large amount of latent heat energy at an appropriate time.

In our F-FSPCMs modules testing, after charging for 5 min and disconnecting the power supply, the module can maintain a constant surface temperature of 39-42 °C for 22-40 min. However, the common direct electric heating temperature control module requires a temperature feedback mechanism to frequently power on and off to maintain a stable temperature. Obviously, the module proposed here does not require more complex temperature control circuits and high-frequency power-on and power-off conditions. Especially, it can be used as a portable module, and most of the time there is no need to connect to a power supply in practical utilization, making it suitable for emergency response filing options.

Fig. R19 Adaptive temperature indication F-FSPCMs temperature control module in future work (unpublished). a Visual colour. **b** Infrared temperature. **c** Colour changing indication.

Furthermore, we develop a color change module for F-FSPCMs to indicate temperature variation in follow-up study, where the phase change temperature is regarded as the color change temperature to achieve self-adapting charging/discharging effects (**Fig. R19**).

As for the energy efficiency of the module, the value is closely related to the module parameters and usage conditions (e.g., thickness of module and heating time). In fact, the results obtained from **Fig. R20** and **Table R3** (**Fig. 6c** and **Table S4** in the revised manuscript) exactly provide reference criteria for optimizing configuration in thermal management applications.

Fig. R20 Temperature holding time of the F-FSPCMs module with the same heat supply for 3 mm, 5 mm and 7 mm thickness.

Table R2 Thermal efficiency of F-FSPCMs (S2) temperature control modules.

Module	Thickness (mm)	Mass (g)	ΔH_f (J/g)	Power (W)	Time (s)	Maximum surface temperature (°C)	η (%)
1	3	31.2	174.4	22	900	79.3	27.5
2	5	52.1	174.4	22	900	62.1	45.9
3	7	72.9	174.4	22	900	54.5	64.2
4	7	72.9	174.4	22	800	50.1	72.2
5	7	72.9	174.4	22	600	43.3	96.3
6	7	72.9	174.4	22	400	37.2	144.5

It is also worth noting that our F-FSPCMs heating method is not limited to high-grade electrical energy, but can also control the temperature of low-grade waste heat. Specially, it has great potential to be used as an insulation and heat absorbing layer for batteries to delay its peak temperature, and thus, improve power generation efficiency. In addition, its excellent hydrophobicity is expected to be used for direct heat exchange of

hot water in industrial waste heat. Even applying our F-FSPCMs as a matrix and incorporating trace amounts of photothermal conversion additives, a "photothermal" conversion composite material can be obtained in this case, allowing a clean energy source of solar energy to be used as one heat source. As a whole, the resultant F-FSPCMs herein exhibit excellent shape stability and the ability to smooth out high-density heat flux during practice, making it worthy of development in numerous application scenarios. Based on your kind comment, we have supplemented relevant statements to improve this part in the revised manuscript.

Changes add to manuscript:

Furthermore, a color change module for F-FSPCMs is developed to indicate temperature variation, where the phase change temperature is used to stimulate color change to achieve self-adapting charging/discharging. Therefore, it avoids the frequent temperature feedback for power on and off in the conventional power-heat supply methods. **(Page 26 Line 479)**

Importantly, the thermal charging of F-FSPCMs module is not limited to electrical energy, it can also be heated by low-grade waste heat or solar energy. Specially, it also has great potentials for battery thermal management, solar photothermal conversion and thermal storage. **(Page 29 Line 544)**

Comment 6: Fig. S8b: The authors tested the leakage of F-FSPCMs, however, the results are not very clear. My suggestion is to put a piece of paper under F-FSPCM, then heat it and observe if PCM leaks to paper. These results would be clearer than the current ones.

Response: Thank you for the valuable comment and suggestion. We originally conducted leakage testing of F-FSPCMs in water considering that water molecules can fully surround F-FSPCMs, which is believed to be a more destructive leakage test under harsh conditions. Especially compared to the constantly emerging carbon-based phase change composites, the host PCMs adsorbed through capillary force adsorption often melt directly into hot water and cannot maintain their shape stability.

Moreover, we conduct 500 heating/cooling cycles (**Fig. R21**, **Fig. 3c** in the revised manuscript), demonstrating the leakage-proof performance of F-FSPCMs. Additionally, it is observed that F-FSPCMs with pronounced leakage are unable to withstand tensile testing and endure compression durability testing above the phase transition temperature. Our consistent mechanical performance data indirectly support this conclusion.

Fig. R21 Enthalpy and phase change temperature of PW@OBC-SEBS (S2) after 500 accelerated thermal cycles.

Undoubtedly, it is a good way to intuitively evaluate the leakage-proof performance by "heating the prepared F-FSPCMs and observing if PCM leaks to paper" as suggested by the reviewer. For better illustrate the results, we perform additional test to heat the F-FSPCMs on a hot plate at a temperature of 80 °C. The photos of the test samples show that paraffin wax has serious leakage while our F-FSPCMs exhibits excellent leakage-proof performance during the heating process, as depicted in **Fig. R22**.

Fig. R22 Leakage comparison diagram of paraffin wax and F-FSPCMs (S2) heated on a hot plate.

Changes add to Supplementary Information (SI):

Fig. R22. is added in SI as Supplementary **Fig. S14**

The disparity in leakage behavior between paraffin wax (PW) and F-FSPCMs is evident when subjected to an 80 °C hot plate covered with filter paper. PW rapidly undergoes complete liquefaction, forming a distinct puddle of liquid, whereas F-FSPCMs exhibit robust shape retention on the filter paper, displaying no observable indications of leakage.

(Page 20 Line 244)

Finally, we like to sincerely thank the reviewers again for your valuable comments and constructive suggestions! We hope you would find this revised version now can be accepted for publication.

References

1. Thitithammawong, A., Nakason, C., Sabakaro, K. & Noordermeer, J. W. M. Thermoplastic vulcanizates based on epoxidized natural rubber/polypropylene blends: Selection of optimal peroxide type and concentration in relation to mixing conditions. *Eur. Polym. J.* **43**, 4008–4018 (2007).
2. Wang, S. *et al.* Facile mechanochemical cycloreversion of polymer cross-linkers enhances tear resistance. *Science* **380**, 1248–1252 (2023).
3. Gao, L. *et al.* Intrinsically elastic polymer ferroelectric by precise slight cross-linking. *Science* **381**, 540–544 (2023).
4. Yang, G., Gong, Z., Luo, X., Chen, L. & Shuai, L. Bonding wood with uncondensed lignins as adhesives. *Nature* **621**, 511–515 (2023).
5. Zhang, Q., Zhao, Y. & Feng, J. Systematic investigation on shape stability of high-efficiency SEBS/paraffin form-stable phase change materials. *Sol. Energy Mater. Sol. Cells* **118**, 54–60 (2013).
6. Chriaa, I. *et al.* Thermal properties of shape-stabilized phase change materials based on low density polyethylene, hexadecane and SEBS for thermal energy storage. *Appl. Therm. Eng.* **171**, 115072 (2020).
7. Xiang, B., Yang, Z. & Zhang, J. ASA/SEBS/paraffin composites as phase change material for potential cooling and heating applications in building. *Polym. Adv. Technol.* **32**, 420–427 (2021).
8. Yang, Y., Cai, X. & Kong, W. A novel intrinsic photothermal and flexible solid-solid phase change materials with super mechanical toughness and multi-recyclability. *Appl. Energy* **332**, 120564 (2023).
9. Li, C., Li, Q., Ge, R. & Lu, X. A novel one-step ultraviolet curing fabrication of myristic acid-resin shape-stabilized composite phase change material for low temperature thermal energy storage. *Chem. Eng. J.* **458**, 141355 (2023).
10. Zhao, X. *et al.* A shape-memory, room-temperature flexible phase change material based on PA/TPEE/EG for battery thermal management. *Chem. Eng. J.* **463**, 142514 (2023).

11. Qi, X., Shao, Y., Wu, H., Yang, J. & Wang, Y. Flexible phase change composite materials with simultaneous light energy storage and light-actuated shape memory capability. *Compos. Sci. Technol.* **181**, 107714 (2019).
12. Yan, Y., Li, W., Zhu, R., Lin, C. & Hufenus, R. Flexible phase change material fiber: A simple route to thermal energy control textiles. *Materials* **14**, 401 (2021).
13. Lin, Y. *et al.* Flexible, highly thermally conductive and electrically insulating phase change materials for advanced thermal management of 5G base stations and thermoelectric generators. *Nano-Micro Lett.* **15**, 31 (2023).
14. Wu, S. *et al.* Highly thermally conductive and flexible phase change composites enabled by polymer/graphite nanoplatelet-based dual networks for efficient thermal management. *J. Mater. Chem. A* **8**, 20011–20020 (2020).
15. Zhang, Q. *et al.* Investigation on the recovery performance of olefin block copolymer/hexadecane form stable phase change materials with shape memory properties. *Sol. Energy Mater. Sol. Cells* **132**, 632–639 (2015).
16. Chen, P. *et al.* Metal foam embedded in SEBS/paraffin/HDPE form-stable PCMs for thermal energy storage. *Sol. Energy Mater. Sol. Cells* **149**, 60–65 (2016).
17. Zhang, Q. *et al.* Polyethylene glycol/polyurethane acrylate-based flexible phase-change film with excellent mechanical strength and reversible optical performance. *Energy Fuels* **37**, 3227–3235 (2023).
18. Huang, Q. *et al.* Pouch lithium battery with a passive thermal management system using form-stable and flexible composite phase change materials. *ACS Appl. Energy Mater.* **4**, 1978–1992 (2021).
19. Deng, C. *et al.* Synchronous visual/infrared stealth using an intrinsically flexible self-healing phase change film. *Adv. Funct. Mater.* **33**, 2212259 (2023).
20. Li, X. *et al.* Wearable janus-type film with integrated all-season active/passive thermal management, thermal camouflage, and ultra-high electromagnetic shielding efficiency tunable by origami process. *Adv. Funct. Mater.* **33**, 2212776 (2023).
21. Çakır, N. Y., İnan, Ö., Ergün, M., Kodal, M. & Özkoç, G. Unlocking the potential use of reactive POSS as a coagent for EPDM/PP-based TPV. *Polymers* **15**, 2267 (2023).

22. Drobny, J. G. Styrenic block copolymers. in *Handbook of Thermoplastic Elastomers (Second Edition)* (ed. Drobny, J. G.) 175–194 (2014).
23. Rodak, A., Susik, A., Kowalkowska-Zedler, D., Zedler, Ł. & Formela, K. Cross-linking, morphology, and physico-mechanical properties of GTR/SBS blends: dicumyl peroxide vs. sulfur system. *Materials* **16**, 2807 (2023).
24. Yu, S. *et al.* Skeletal network enabling new-generation thermoplastic vulcanizates. *Adv. Mater.* **35**, 2300856 (2023).
25. Dlużneski, P. R. Peroxide vulcanization of elastomers. *Rubber Chem. Technol.* **74**, 451–492 (2001).
26. Li, Z. *et al.* Supramolecular and physically double-cross-linked network strategy toward strong and tough elastic fibers. *ACS Macro Lett.* **9**, 1655–1661 (2020).
27. Lu, Y. *et al.* Magnetically tightened form-stable phase change materials with modular assembly and geometric conformality features. *Nat. Commun.* **13**, 1397 (2022).
28. Lepage, M. L. *et al.* A broadly applicable cross-linker for aliphatic polymers containing C–H bonds. *Science* **366**, 875–878 (2019).

REVIEWER COMMENTS

Reviewer #1 (Remarks to the Author):

In the revised manuscript, the authors have made a lot of modifications and supplemented many experiments. Moreover, the authors further elaborate on the innovation of this work. In my opinion, this paper is worth publishing in Nature Communications.

Reviewer #2 (Remarks to the Author):

The authors have responded my previous comments effectively; our six queries have been resolved appropriately. I would therefore recommend this manuscript can be considered for publication.

Responses to Referee #1

In the revised manuscript, the authors have made a lot of modifications and supplemented many experiments. Moreover, the authors further elaborate on the innovation of this work. In my opinion, this paper is worth publishing in Nature Communications.

Response: We are pleased that the reviewer acknowledged the novelty and significance of our research. We truly believe that the reviewers' comments improved significantly our work. Thank you for recommending our revised manuscript to be published in Nature Communications.

Responses to Referee #2

The authors have responded my previous comments effectively; our six queries have been resolved appropriately. I would therefore recommend this manuscript can be considered for publication.

Response: We appreciate the reviewer for careful reading of the manuscript and the specific comments. We are also grateful for the reviewer's affirmation on our revised manuscript.